# New Definition of *Neoprotereunetes* Fain et Camerik, Its Distribution and Description of the New Genus in Eupodidae (Acariformes: Prostigmata: Eupodoidea) [note 1]

**DOI:** 10.3390/ani13132213

**Published:** 2023-07-05

**Authors:** Ronald Laniecki, Wojciech L. Magowski

**Affiliations:** Department of Animal Taxonomy and Ecology, Adam Mickiewicz University, Uniwersytetu Poznańskiego 6, 61-614 Poznań, Poland; ronald.laniecki@amu.edu.pl or ronlaniecki@gmail.com

**Keywords:** Acari, *Protereunetes*, taxonomy, biogeography, polar regions

## Abstract

**Simple Summary:**

The mite genus *Neoprotereunetes*, long neglected in the literature, is revised according to modern taxonomic standards. Six species from both Arctic and Antarctic locations, previously placed in the genera *Protereunetes* or *Eupodes*, are transferred to *Neoprotereunetes*. The new genus *Antarcteupodes* is created to accommodate one Antarctic species *A. maudae* comb. nov, originally described in *Protereunetes*. An identification key to *Neoprotereunetes* is provided.

**Abstract:**

The genus *Neoprotereunetes* Fain et Camerik, 1994 is revised and its definition is extended in order to incorporate some species of the invalid genus *Protereunetes* Berlese, 1923. The former type species *Neoprotereunetes—Ereunetes lapidarius* Oudemans, 1906 is redescribed and transferred to *Filieupodes* Jesionowska, 2010 (Cocceupodidae); *Proterunetes boerneri* is redescribed and designated the new type species. Two species groups are proposed to embrace Arctic and Antarctic species, respectively. *Protereunetes paulinae* Gless, 1972 is redescribed, whereas *Protereunetes maudae* Strandtmann, 1967 is redescribed and designated the type species of the new genus *Antarcteupodes* gen. nov. A key to the species of *Neopretereunetes* is provided.

## 1. Introduction

Superfamily Eupodoidea C.L. Koch, 1842 gathers mostly cosmopolitan, terrestrial, soft-bodied and often-colorful mites. Most of them are mycophagous, but there are also predacious (Rhagidiidae) and phytophagous groups (Penthaleidae and Penthalodidae). Some of them, like *Penthaleus major* (Dugès, 1834) (Penthaleidae) and *Halotydeus destructor* (Tucker, 1925) (Penthalodidae), are significant crop pests, whereas *Linopodes* sp. (Cocceupodidae) is considered an economic pest in mushroom houses [1]. Eupodoidea is divided into nine families: Eupodidae C.L. Koch, 1842; Rhagidiidae Oudemans, 1922; Penthaleidae Oudemans, 1931; Penthalodidae Thor, 1933; Strandtmanniidae Zacharda, 1979; Eriorhynchidae Qin et Halliday, 1997; Pentapalpidae Olivier et Theron, 2000; Dendrochaetidae Olivier, 2008 and Cocceupodidae Jesionowska, 2010 [2]. However, internal relationships among families within Eupodoidea remain uncertain [3].

Family Eupodidae C.L. Koch, 1842 currently includes 11 genera: *Eupodes* C.L. Koch, 1835; *Benoinyssus* Fain, 1958; *Claveupodes* Strandtmann et Prasse, 1976; *Caleupodes* Baker, 1987; *Niveupodes* Barillo, 1991; *Neoprotereunetes* Fain et Camerik, 1994; *Aethosolenia* Baker et Lindquist, 2002; *Xerophiles* Jesionowska, 2003; *Pseudoeupodes* Khaustov, 2014; *Pseudopenthaleus* Khaustov, 2015 and *Echinoeupodes* Khaustov, 2017. Genera *Linopodes* Koch, 1835 and *Cocceupodes* Thor, 1934 (previously in Eupodidae) along with one new genus *Filieupodes* (Jesionowska, 2010), were placed by Jesionowska [4] in the separate family Cocceupodidae Jesionowska, 2010, still within Eupodoidea.

Representatives of the genus *Neoprotereunetes* (as diagnosed herewith) are small, inconspicuous mites. Their bodies are pale white often with dark- to light-green-colored idiosoma, divided by a white medial longitudinal stripe, that apparently being an intestine, showing through the lucent integument. Two pigment eye spots occur on the prodorsum but do not preserve in permanent microscopic slides. These fast-moving mites inhabit soil, mosses, lichens, grasses and mammal nests, and have been observed feeding on algae [5].

The history of the genus is long and complex. The subgenus *Protereunetes* was erected within the genus *Micrereunetes* (Tydeoidea: Ereynetidae) by Berlese [6], with the type species *M.* (*P.*) *agilis* and another species, *M.* (*P.*) *brevipes*. Thor [7] raised *Protereunetes* to the generic rank and placed it in the family Eupodidae. Next, Thor and Willmann [8] included five species in *Protereunetes*: *P. striatellus* (C.L. Koch, 1838), *P. lapidarius* (Oudemans, 1906), *P. agilis* (Berlese, 1923), *P. brevipes* (Berlese, 1923) and *P. börneri* Thor, 1934. Subsequently, Fain [9] redescribed *P. agilis* and *P. brevipes*, showing them actually belonging in the genus *Ereynetes* Berlese, 1883 (Tydeoidea: Ereynetidae), meaning that *Protereunetes* is a junior synonym of *Ereynetes*. Regardless of that, in the next twenty years, some new eupodid species were still described in *Protereunetes*: *P. minutus* Strandtmann, 1967; *P. maudae* Strandtmann, 1967; *P. crozeti* Strandtmann et Davies, 1972 and *P. paulinae* Gless, 1972. Although Strandtmann [10], noticing the results of Fain’s study [9], transferred *P. minutus* to the genus *Eupodes*, he did not sustain his own view in subsequent papers [11,12]. This new combination, however, was widely accepted by subsequent authors, e.g., Goddard [13], Booth et al. [14], Baker [15]. Lastly, Fain and Camerik [16] created a new genus—*Neoprotereunetes* with the type species *Neoprotereunetes lapidarius* (Oudemans, 1906)—to bracket those species, which were described in *Protereunetes* until then and, unlike *P. agilis* and *P. brevipes*, belonged to the family Eupodidae. However, Fain and Camerik did not present any firm diagnosis for the new genus and only pointed at inaccurate and outdated definition of *Protereunetes* of Thor and Willmann [8]. Moreover, thanks to the present study, their designated type species, *Neoprotereunetes lapidarius* (Oudemans, 1906) appears to be a senior synonym of *Filieupodes filistellatus* Jesionowska, 2010, from the family Cocceupodidae (Eupodoidea). *Neoprotereunetes* as a genus-level taxon appeared in the literature only once more, in the revision of the family Eupodidae by Khaustov [17]. Thus, the aims of the present study are: (1) to redescribe *Ereunetes lapidarius* and correct its systematic position; (2) to provide new definition for the genus *Neoprotereunetes*; (3) to list species in this genus according to the revised diagnosis; (4) to designate its new type species; and (5) to create an identification key for the species within the genus.

## 2. Material and Methods

The material of *Neoprotereunetes boerneri* was extracted from soil samples using a Berlese–Tullgren funnel (photo-eclector) for one day and stored in 75% ethanol. Thereafter, specimens were cleared in lactic acid, mounted in Hoyer’s medium on glass slides and heated for 10–15 days at the temperature of 55 °C. The type material of *Eupodes minutus*, *Protereunetes maudae* and *Protereunetes paulinae* was loaned from collection at Bishop Museum in Honolulu (Hawaii) and the type material of *Neoprotereunetes lapidarius* was loaned from collection at Naturalis Biodiversity Center in Leiden (the Netherlands) (Figure 1). Mites were studied with a phase contrast (PC) (Olympus BX41, BX50) and differential interference contrast (DIC) (BX51) microscopes and identified using keys of Booth et al. [14], Jesionowska [4] and Khaustov [17], as well as original descriptions. Measurements were obtained from the specimens with the aid of an ocular micrometer and are given in micrometers (μm). The drawings were performed using a drawing tube (camera lucida) and processed in the Corel PHOTO-PAINT X5 program. Micrographs were taken using a Canon D5 Mk. II DSLR camera; pictures were assembled and processed with PICOLAY stacking software [18] or manually in Corel PHOTO-PAINT X5.

Morphological nomenclature for idiosoma and gnathosoma follows Baker and Lindquist [3]; for leg chaetotaxy, a universal Grandjean’s notation system, reviewed by Norton [19] and applied for eupodoids by Lindquist and Zacharda [20] and Baker [21], is used. The spine-like seta on tibia I is treated herein as a famulus, and thus designated by the Greek letter kappa (*κ*) rather than the Latin letter *k*, analogically to the famuli on tarsi I and II designated by the Greek letter epsilon (*ε*). Palpal and leg setal formulae are given from trochanters to tarsi with solenidia and famuli indicated in parentheses. The setae for basi- and telofemora are given separately, even when segment is not divided. The terms “long” and “short” related to dorsal hysterosomal setae mean values equal to or longer than the distance between members of a pair of setae and shorter than this distance, respectively. This excludes lateral hysterosomal setae, i.e., *c*_2_, *f*_2_ and *h*_2_, and also setae *f*_1_ and *h*_1_, which are more tightly clustered at rear part of hysterosoma (caudal bent). Those are longer than remaining hysterosomal setae, and the latter in some eupodid genera (e.g., *Benoinyssus*, *Aethosolenia*) differentiated into trichobothria. Eupathidia are treated herein as setae (1) completely hollow, and (2) with a widely open base and designated by the Greek letter zeta (ζ), subtending the name of a seta. When a seta does not fulfill both conditions (e.g., it is partially hollow) it is then designated by “ζ?”. Abbreviations used: *ap*—subcapitular apodema, *cpc*—podocephalic canal, *LL*—lateral lip, *LS*—labrum, *OE*—esophagus, *tr*?—trachea?. Diagnoses and descriptions of taxa refer to adult females if not stated otherwise.

## 3. Results

### 3.1. Systematics

Superfamily Eupodoidea C.L. Koch, 1842

Family Eupodidae C.L. Koch, 1842


***Neoprotereunetes* Fain et Camerik, 1994**


Type species: *Protereunetes boerneri* Thor, 1934 by new designation.

Diagnosis. Sejugal furrow present. Idiosomal integument with striate-spiculate ornamentation. Internal vertical setae (*v_1_*) inserted in common areolae, on well-delimited naso. Prodorsal trichobothria (*sc_1_*) filiform and pilose. Hysterosomal setae short and pilose. No hysterosomal trichobothria. Coxisternal setal formula: 3–1–4–3. Six (or exceptionally five) pairs of genital setae in single row and none more lateral than others. Three pairs of pseudanal setae. Adanal setae absent. Four pairs of lyrifissures. Palpal setal formula: 0–2–3–9(*ω*). All legs shorter than body. Femora IV not enlarged. Leg integument with spiculate ornamentation. Tibiae I and II each with two rhagidial organs.

Description. Idiosomal dorsum. Sejugal furrow present. Integument with striate-spiculate ornamentation. Prodorsum bearing four pairs of setae: internal verticals (*v_1_*), external verticals (*v_2_*), internal scapulars (*sc_1_*) and external scapulars (*sc_2_*). Naso basally delimited from prodorsal shield and bearing setae *v_1_*. Setae *sc_1_* trichobothrial, short, not reaching the posterior edge of naso, pilose. Remaining prodorsal setae short, pilose and inserted in typical areolae. Hysterosoma bearing eight pairs of dorsal setae: internal humerals (*c_1_*), external humerals (*c_2_*), first dorsals (*d_1_*), second dorsals (*e_1_*), internal lumbars (*f_1_*), external lumbars (*f_2_*), internal sacrals (*h_1_*) and external sacrals (*h_2_*). All hysterosomal setae short, pilose, inserted in typical areolae and none of them trichobothrial. Three pairs of dorsal lyrifissures (*ia*, *im*, *ip*) present.

Idiosomal venter. Coxisternal fields integument with weakly striate-spiculate ornamentation. Coxisternal setal formula: 3–1–4–3. Small cavities near outer margin of coxae I-III present. Genital aperture posteroventral, flanked by four or five pairs of aggenital setae (*ag_1_*_-*4* or -*5*_). Genital valves bearing six (or exceptionally five) pairs of genital setae (*g_1_*_-*6* (-*5*)_) of which the anterior first is longer than the second and the second is longer than the remaining ones. All setae *g* are always in single row and none more lateral than others. Internal genital structures consist of two pairs of genital papillae and four or six pairs of eugenital setae (*eu_1_*_-*4* or -*6*_) set on protuberances. Anal opening terminal, flanked by three pairs of pseudanal setae: *ps_1_*_,*2*_ posteriorly (sometimes located terminally or dorsally) and shorter *ps_3_* anteriorly. No anal setae (*an*) on anal valves. One pair of ventral lyrifissures (*ih*) present.

Gnathosoma. Subcapitulum roughly triangular, bearing four pairs of setae: two pairs of pilose subcapitular setae (*sbc_1_*_,*2*_) and two pairs of minute smooth adoral setae (*or_1_*_, *2*_). Setae *sbc_1_* usually thinner and shorter than *sbc_2_*, both located along the base of each lateral lip, at antiaxial and paraxial end of subcapitular apodema, respectively. Setae *or_1_* and *or_2_* closely clustered at the tip of each lateral lip and often hard to discern. Apex of labrum acuminate. Chelicerae slender, bearing short, smooth dorsal seta *cha*. Palps four-segmented with weakly barbed supracoxal seta *ep*. Palpal setal formula: 0–2–3–9(*ω*). Tarsus laterally flattened, bearing nine setae: dorsal (*d*), two laterals (*l*′, *l*″), sublateral (*sl*″), anteroculminal (*acm*), two prorals (*p*′, *p*″), ventral (*v*), basal (*ba*) and small rhagidial organ *ω*.

Legs. Legs I and IV longer than II and III, but all shorter than body. Femora I subdivided ventrally, II undivided, III and IV divided. All apoteles with ambulacra, consisting of pad-like empodium with dense setulae arranged in bands on lateral margins and pair of hooked claws with short outgrows on its ventral surface. Integument with spiculate ornamentation. All setae densely pilose except for sparsely pilose *v*′ on trochanters I and II and weakly barbed supracoxal seta *el*. Solenidia and famuli. Leg I. Genu with one dorsomedial erect solenidion *σ*. Tibia with one anterior complex of rhagidial organ *φ_1_* and spiniform famulus *κ*, and one medial rhagidial organ *φ_2_*, tandemly or obliquely in separated depressions. Tibial rhagidial organs long and T-shaped or L-shaped, or either short and ellipsoid to almost spherical. Tarsus with two rhagidial organs *ω_1_*_,_*_2_* and one stellate famulus *ɛ*, in tandem in confluent or separate depressions. Posterior one two to three times longer than anterior one. Leg II. Genu with or without medial, erect solenidion *σ*. Tibia with two rhagidial organs *φ_1_*_,*2*_ (anterior and medial), tandemly in separate depressions. Tarsus with two or three rhagidial organs and with or without spiniform famulus *ɛ*, variously arranged. Mostly three rhagidial organs present, in confluent depression arranged alternately, i.e., anterior and posterior rhagidial organs situated antiaxially, whereas the medial one is situated paraxially. However, only two rhagidial organs can be present and situated obliquely in separate depressions or in tandem in confluent depression. Leg III. Genu without solenidion. Tibia with or without proximal rhagidial organ. Tarsus without rhagidial organs. Leg IV without solenidia.

Differential diagnosis. The genus resembles *Caleupodes* Baker, 1987 in having short dorsal setae, all legs shorter than the body, femora IV not enlarged and two rhagidial organs on both tibiae I and II. It differs from *Caleupodes* in having integument with striate-spiculate ornamentation (reticulate in *Caleupodes*), pilose dorsal setae (weakly serrate in *Caleupodes*), six or five genital setae (seven in *Caleupodes*) and three pairs of pseudanal setae (two in *Caleupodes*). *Neoprotereunetes* also shares some similarities with the genus *Pseudoeupodes* Khaustov, 2014, i.e., short dorsal setae, legs shorter than body and femora IV not enlarged. It differs from *Pseudoeupodes* in having five or six genital setae (six in *Pseudoeupodes*), three pairs of pseudanal setae (two in *Pseudoeupodes*), two rhagidial organs on both tibiae I and II (one rhagidial organ and one erect solenidion in *Pseudoeupodes*).

Species belonging to the genus *Neoprotereunetes*:*Protereunetes boerneri* Thor, 1934*Protereunetes crozeti* Strandtmann et Davies, 1972*Eupodes exiguus* Booth, Edwards et Usher, 1985*Eupodes minutus* (Strandtmann, 1967)*Eupodes parvus* Booth, Edwards et Usher, 1985*Protereunetes paulinae* Gless, 1972

*Neoprotereunetes boerneri* species group

Diagnosis. Genital region with five aggenital, six genital and six eugenital setae. Tarsus I with 21 setae (additional ventro-lateral antiaxial seta on tarsus I between setae *pv*″ and *v_1_*″). Tarsus IV with 13 setae. Tibia I with five setae. Genua III and IV each with four setae. Femur I with 13 setae. Arctic and sub-Arctic distribution. Currently the group contains only one species (*N. boerneri*).

***Neoprotereunetes boerneri* (Thor, 1934)** comb. nov. (Figure 2, Figure 3, Figure 4, Figure 5, Figure 6 and Figure 7)

*Protereunetes börneri* [7,8]

*Protereunetes boerneri* [11,22]

*Protereynetes boerneri* (sic!) [23]

*Neoprotereunetes borneri* (sic!) [24]

Diagnosis. Genital region with five pairs of *ag* and six pairs of *g* setae. An extra ventro-lateral, antiaxial seta on tarsus I, located between setae *pv*″ and *v_1_*″. Trochanter IV with one seta. Two rhagidial organs on tarsus II, slantwise in separated depressions. Proximal rhagidial organ on tibia I and II long, at most four times shorter than its segment.

Redescription. Female. Idiosoma 220 long, 113 wide.

Idiosomal dorsum (Figure 2 and Figure 7A). Prodorsal shield 57 long, 82 wide, triangular. Prodorsal integument with weakly striate-spiculate ornamentation, but course of striae hard to retrace. Naso (Figure 2 and Figure 7B) 9 long, 13 wide, rounded. Lengths of prodorsal setae: *v_1_* 9, *v_2_* 17, *sc_1_* ca. 40, *sc_2_* 16; distances: *v_1_*–*v_1_* 3, *v_2_*–*v_2_* 44, *sc_1_*–*sc_1_* 29, *sc_1_*–*sc_1_* 64. Hysterosoma tapering caudally, its frontal corners protruding laterally over prodorsum. Hysterosomal integument with striate-spiculate ornamentation. Lengths of hysterosomal setae: *c_1_* 11, *c_2_* 19, *d_1_* 11, *e_1_* 10, *f_1_* 12, *f_2_* 19, *h_1_* 17, *h_2_* 17; distances: *c_1_*–*c_1_* 22, *c_1_*–*c_2_* 44, *d_1_*–*d_1_* 37, *e_1_*–*e_1_* 24, *f_1_*–*f_1_* 20, *f_1_*–*f_2_* 20, *h_1_*–*h_1_* 12, *h_1_*–*h_2_* ca. 14.

Idiosomal venter (Figure 3 and Figure 7C). Coxisternal fields outlined, with weakly striate-spiculate ornamentation, separated medially by striate-spiculate ornamentation of longitudinal course. Lengths of coxisternal setae: *1a* 12, *1b* 14, *1c* 9, *2b* 11, *3a* 10, *3b* 11, *3c* 11, *3d* 11, *4a* 9, *4b* 10, *4c* 10; distances: *1a*–*1a* 19, *1b*–*1b* 40, *1c*–*1c* 65, *2b*–*2b* 66, *3a*–*3a* 15, *3b*–*3b* 80, *3c*–*3c* 90, *3d*–*3d* 61, *4a*–*4a* 20, *4b*–*4b* 81, *4c*–*4c* 57. Coxal cavities well defined. Genital region (Figure 4A) with five pairs of aggenital setae: *ag_1_* 8 long, *ag_2_*_-*5*_ 7 long, and six pairs and genital setae: *g_1_* 8 long, *g_2_* 7 long, *g_3_*_-*6*_ 6 long. Six pairs of eugenital setae, ca. 6 long, on protuberances (Figure 4B). Sternal (*1a*, *3a*, *4a*), genital, aggenital and *ps_3_* slightly expanded distally. Lengths of pseudanal setae: *ps_1_* 14, *ps_2_* 16, *ps_3_* 10.

Gnathosoma (Figure 4C–G and Figure 7D,E). Subcapitulum (Figure 4C) 66 long, 31 wide, slender, roughly triangular, with spiculate ornamentation. Subcapitular apodema not visible. Setae *sbc_2_* 8 long, densely pilose, thicker than sparsely pilose *sbc_1_*, 3 long. Chelicerae (Figure 4D) 52 long, 15 wide, with spiculate ornamentation, bearing small smooth dorsal seta *cha*. Fixed digit with two pointed tips, ventral pointing forward and dorsal slightly curved backward; movable digit sharp, clawlike. Palps (Figure 4E–G and Figure 7D,E) with spiculate ornamentation, spiculate-cuspidate on femorogenu. Palpal femorogenu 30 long, with indication of division dorsolaterally (Figure 4E). Palp tibial seta *l*′ nearly twice as long as *l*″. Palpal tarsus 18 long, oval in lateral aspect, triangular in dorsoventral aspect. Setae *d*, *l*′, *l*″, *v* and *ba* pilose; setae *sl*″, *acm*, *p*′ and *p*″ smooth; rhagidial organ *ω* ellipsoid with proximal stock.

Legs (Figure 5, Figure 6 and Figure 7F–H). Lengths of legs: I 164, II 94, III 108, IV 140. Lengths of leg segments: I: Ts: 47, Tb 30, G 30, F 64, Tr 27; II: Ts 33, Tb 22, G 18, F 40, Tr 22; III: Ts 34, Tb 23, G 19, TF 15, BF 31, Tr 22; IV: Ts 36, Tb 29, G 24, TF 16, BF 39, Tr 29. Integument with spiculate ornamentation, spiculate-cuspidate on basifemur III and from tibia to trochanter of leg IV. Leg setal formulae: I: 1–8+5–6(*σ*)–5(2*φ*, *κ*)–21(2*ω*, *ɛ*); II: 1–5+5–4(*σ*)–5(2*φ*)–13(2*ω*, *ɛ*); III: 1–4+4–4–5(*φ*)–12; IV: 1–3+3–4–5–13. Leg eupathidial setae: I: Tb: all except *v*′; Ts: all except (*v_3_*) II: Tb: *d*, *l*″; Ts: all except *tc*″ and *it*″; III: Tb: *d*, *v*′; Ts: *it*′, (*p*); IV: G: *l*′; Tb: *d*, *l*′, *v*′; Ts: *tc*. Solenidia and famuli. Leg I. Genu with one dorsomedial erect solenidion *σ*. Tibia with one anterior complex of L-shaped rhagidial organ *φ_1_*, 6 long, plunge into integument basally and spiniform famulus *κ*, and one medial T-shaped rhagidial organ *φ_2_*, 11 long, obliquely in separated depressions. Tarsus with two rhagidial organs and one stellate famulus *ɛ*, obliquely in confluent depression. Posterior one (*ω_1_*) T-shaped, 10 long and anterior one (*ω_2_*) L-shaped, 4 long. Leg II. Genu with one dorsomedial erect solenidion *σ*. Tibia with two rhagidial organs (L-shaped *φ_1_*, 4 long and T-shaped *φ_2_*, 6 long), tandemly in separated depressions. Tarsus with two rhagidial organs (T-shaped *ω_1_*, 10 long and L-shaped *ω_2_*, 4 long), obliquely in separated depressions. Posterior one (*ω_1_*) subtended by spiniform famulus *ɛ*. Leg III. Tibia with proximal T-shaped rhagidial organ *φ*, 6 long.

Tritonymph. Body length 214. Four and three pairs of *ag* and *g* setae, respectively; *eu* setae absent. Leg setal formulae: I: 1–6+5–6(*σ*)–5(2*φ*, *κ*)–18(2*ω*, *ɛ*); II: 1–5+5–4(*σ*)–5(2*φ*)–12(2*ω*, *ɛ*); III: 1–4+4–4–5(*φ*)–10; IV: 1–3+3–4–5–11. Other characters as in adults.

Deutonymph. Body length 198. Coxisternal setal formula: 3–1–3–2. Two pairs of both *ag* and *g* setae; *eu* setae absent. Leg setal formulae: I: 1–5+5–4(*σ*)–5(2*φ*, *κ*)–17(2*ω*, *ɛ*); II: 1–3+5–4(*σ*)–5(2*φ*)–12(2*ω*, *ɛ*); III: 1–2+4–4–4(*φ*)–10/11; IV: 0–1+3–4–4–11. Other characters as in adults.

For male, protonymph and larva see [11].

Differential diagnosis. *N. boerneri* resembles *N. parvus* by presence of two rhagidial organs on tarsus II. It differs from *N. parvus* in having five pairs of *ag* setae (four in *N. parvus*), one seta on trochanter IV (lacking in *N. parvus*) and long proximal rhagidial organs on tibiae I-III (short on tibia I and II, and lacking on tibia III in *N. parvus*).

Distribution. Temple Bay, “Grosser Trichter”, Magdalena Bay, Spitsbergen, Svalbard, Norway [7]; Utqiaġvik (formerly Barrow), Anaktuvuk Pass, Wainwright, Alaska, USA [11]; Bolshevik Island, Severnaya Zemlya, Russia [22].

Material examined. Four females, one tritonymph and one deutonymph: Svalbard, Spitsbergen, mountain slope, NW exposition, 150 m a.s.l., 78°14′08″ N 15°20′05″ E, soil in vicinity of little auk (*Alle alle*) rookery, 17 July 2022, leg. K. Zawierucha, M. Zacharyasiewicz, M. Jastrzębski.

Remarks. The original description lacks some valid diagnostic characters and thus the species is redescribed herewith. The type material of *N. boerneri* does not exist ([25], 408 p.; correspondence with Dr. Vladimir Gusarov, Natural History Museum, University of Oslo), but the specimens collected from Spitsbergen fully fit the original description and figures by Thor [7].

The species was redescribed by Strandtmann [11] on the basis of specimens collected from tundra and from the nests of brown lemming (*Lemmus trimucronatus*) in Alaska. The Strandtmann’s material was not available for this study, but no significant differences between the specimens from Alaska and those from Svalbard were found.

*N. boerneri* possesses a unique character, i.e., one extra ventro-lateral, antiaxial seta on tarsus I, located between setae *pv*″ and *v_1_*″. An additional tarsal seta is present in yet another eupodid species, *Echinoeupodes echinus* Khaustov, 2017. In that species additional seta (designted as “*vs*” by Khaustov [26]) is situated ventrally, between the pair of *pv* setae, and occurs on tarsi of all four legs. As it is hard to determine whether these two cases deal with homologous setae, the extra seta is marked only with an asterisk (*) in our study (Figure 5B and Figure 7G).

*Neoprotereunetes minutus* species group

Diagnosis. Genital region with four aggenital, six (or exceptionally five) genital and four eugenital setae. Tarsus I with 20 setae. Tarsus IV with 11 setae. Tibia I with four setae. Genu III with two or three setae. Genu IV with three setae. Femur I with 12 setae. Antarctic and sub-Antarctic distribution.

***Neoprotereunetes crozeti* (Strandtmann et Davies, 1972)** comb. nov.

*Protereunetes crozeti* [12,22,27]

Diagnosis. Genital region with four pairs of *ag* and six pairs of *g* setae. Trochanter IV with one seta. Tarsus II with three rhagidial organs and without spiniform famulus. Both tarsal rhagidial organs in separate depressions. Proximal rhagidial organs on tibia I and II long, at most four times shorter than its segment. No rhagidial organs on tibia III.

Differential diagnosis. *N. crozeti* resembles *N. minutus* by long proximal rhagidial organs on tibiae and lack of famulus on tarsus II. It differs from *N. minutus* in lacking proximal rhagidial organ on tibia III (present in *N. minutus*) and in arrangement of tarsal rhagidial organs. On tarsus I, in *N. crozeti* tip of antiaxial *ω_1_* and base of paraxial *ω_2_* overlap, whereas in *N. minutus* both are situated medially in tandem. On tarsus II, in *N. crozeti ω_2_* and *ω_3_* lie side by side and in *N. minutus ω_3_* is displaced anteriorly in relation to *ω_2_*.

Distribution. Possession Island, Crozet Islands, ATF [12].

Material examined. None.

Remarks. The original description lacks some valid diagnostic characters, but as the type- or any other material was not available for this study, only standardized diagnosis is given. There is no information on type material deposition in the original paper. It is not deposited in Bishop Museum (courtesy of Dr. Jeremy Frank, Entomology Collections Manager at Bishop Museum).

***Neoprotereunetes exiguus* (Booth, Edwards et Usher, 1985)** comb. nov.

*Eupodes exiguus* [14,22,27]

Diagnosis. Genital region with four pairs of *ag* and six pairs of *g* setae. Trochanter IV with one seta. Tarsus II with three rhagidial organs and spiniform famulus. Both tarsal rhagidial organs in confluent depressions. Proximal rhagidial organs on tibiae I–III short, at least seven times shorter than their segment.

Differential diagnosis. *N. exiguus* resembles *N. parvus* by very short, globular proximal rhagidial organs on tibiae, and T-shaped rhagidial organs on tarsi I and II. It differs from *N. parvus* in number of rhagidial organs on tarsus II (two instead of three) as well as in presence of rhagidial organ on tibia III and seta on trochanter IV (both absent in *N. parvus*).

Distribution. Signy Island, South Orkney Islands [14]; South Shetland Islands [28].

Material examined. One female and one male: King George Island, South Shetland Islands, 62°05′00″ S, 58°23′28″ W, Grasses near the Comandante Ferraz Antarctic Station, 8 February 2016, leg. D.J. Gwiazdowicz.

Remarks. The original description contains all valid diagnostic characters and thus only standardized diagnosis is given.

***Neoprotereunetes minutus* (Strandtmann, 1967)** comb. nov.

*Protereunetes minutus* [29]

*Eupodes minutus* [10,13,14,22,27,30,31,32]

Diagnosis. Genital region with four pairs of *ag* and six pairs of *g* setae. Trochanter IV with one seta. Tarsus II with three rhagidial organs and without spiniform famulus. Both tarsal rhagidial organs in confluent depressions. Proximal rhagidial organs on tibiae I and II at most four times shorter than their segment.

Differential diagnosis. *N. minutus* closely resembles *N. crozeti*, by long proximal rhagidial organs on tibiae and lack of famulus on tarsus II. *N. minutus*, however, possess a proximal rhagidial organ on tibia III (lacking in *N. crozeti*). Additionally, the arrangement and shape of rhagidial organs is different in these two species. On tarsus I, in *N. minutus ω_1_* and *ω_2_* lie parallel, while in *N. crozeti* they lie in tandem. On tarsus II, in *N. minutus ω_3_* is displaced anteriorly in relation to *ω_2_* and in *N. crozeti ω_2_* and *ω_3_* lie side by side.

Distribution. Anvers Island, Palmer Archipelago [29]; Signy Island, South Orkney Islands [14]; Dunedin, New Zeland [30]; Marion Island, Prince Edward Islands, South Africa [31]; King George Island, Halfmoon Island, Deception Island, South Shetland Islands [28].

Material examined. Holotype male (Bishop Museum, slide labeled “BBM 7055”): Antarctic Peninsula, Anvers Island, Norsel Point, 64°30′ S 63°30′ W, under stones and mosses, March 17. 1965, Coll. D. Strong; one female: King George Island, South Shetland Islands, 62°05′00″ S, 58°23′28″ W, Grasses near the Comandante Ferraz Antarctic Station, 8 February 2016, leg. D.J. Gwiazdowicz.

Remarks. The redescription by Booth et al. [14] contains all valid diagnostic characters and thus only a standardized diagnosis is given here. 

Except the type locality, records published before 1985 are not included as suggested in [14].

Mites collected from subalpine grasslands of Mt. Aso and Mt. Kamegamori in Japan were identified by Shiba [33] as *P. minutus*. However, the depicted specimen does not fully agree with the original description and figures as well as the holotype of *P. minutus*. It has shorter rhagidial organs on tibiae I and II and shows rather unusual solenidiotaxy of tibia II (two rhagidial organs and one erect solenidion; see [33], Figure 7e), which does not occur in any other eupodoid species. As the solenidiotaxy of tibiae is not commented in the description and thus cannot be confronted with that figure, this record remains dubious.

Luxton [30] recorded *N. minutus* from Dunedin, New Zeland and refered this species to *Eupodes antipodus* (Womersley, 1937). As no nomenclatorial act was established or synonymy commented it is not included here.

***Neoprotereunetes parvus* (Booth, Edwards et Usher, 1985)** comb. nov.

*Eupodes parvus* [14,22,27]

Diagnosis. Genital region with four pairs of *ag* and six pairs of *g* setae. Tarsus II with two rhagidial organs and spiniform famulus. Both tarsal rhagidial organs in confluent depressions. Proximal rhagidial organs on tibia I and II short, at least seven times shorter than its segment. No rhagidial organs on tibia III. Trochanter IV without setae.

Differential diagnosis. *N. parvus* resembles *N. exiguus* by short proximal rhagidial organs on tibiae and T-shaped rhagidial organs on tarsi I and II. It differs from *N. exiguus* in number of rhagidial organs on tarsus II (two instead of three) as well as in absence of rhagidial organ on tibia III and seta on trochanter IV (both present in *N. exiguus*).

Distribution. Signy Island, South Orkney Islands, South Shetland Islands, Antarctic Peninsula [14]; King George Island and Ardley Island [28].

Material examined. One female and one male: King George Island, South Shetlands, the Antarctic, 62°09′49″ S 58°27′57″ W, nest of the south polar skua (*S. maccormicki*), 27, 28, 31 January 2016, leg. D.J. Gwiazdowicz.

Remarks. The original description contains all valid diagnostic characters, and therefore only a standardized diagnosis is given here.

Two subspecies of *N. parvus* were proposed by Booth et al. [14]: *N. parvus parvus* from South Orkney Islands and *N. parvus grahamensis* from South Shetland Islands and Antarctic Peninsula, which differs from nominative subspecies only in body length and lengths of idiosomal setae (see [14]).

***Neoprotereunetes paulinae* (Gless, 1972)** comb. nov. (Figure 8, Figure 9, Figure 10, Figure 11, Figure 12 and Figure 13)

*Protereunetes paulinae* [5,22,27]

Diagnosis. Genital region with four pairs of *ag* and five pairs of *g* setae. Trochanter IV without setae. Tarsus II with three rhagidial organs and spiniform famulus. Rhagidial organs on tarsi I and II in confluent depression. Proximal rhagidial organs on tibia I and II short, at least seven times shorter than its segment.

Redescription. Holotype female. Idiosoma 268 long, 178 wide.

Idiosomal dorsum (Figure 8 and Figure 13A). Prodorsal shield (Figure 13B) 64 long, 80 wide, triangular. Prodorsal integument with weakly striate-spiculate ornamentation, but course of striae hard to retrace. Naso 10 long, 20 wide, rounded. Lengths of prodorsal setae: *v_1_* 11, *v_2_* 20, *sc_1_* ca. 47, *sc_2_* 19; distances: *v_1_*–*v_1_* 3, *v_2_*–*v_2_* 49, *sc_1_*–*sc_1_* 34, *sc_2_*–*sc_2_* 73. Hysterosoma tapering caudally, its frontal corners protruding laterally over prodorsum. Lengths of hysterosomal setae: *c_1_* 16, *c_2_* 21, *d_1_* 16, *e_1_* 15, *f_1_* 18, *f_2_* 22, *h_1_* 24, *h_2_* 23; distances: *c_1_*–*c_1_* 26, *c_1_*–*c_2_* 65, *d_1_*–*d_1_* 42, *e_1_*–*e_1_* 41, *f_1_*–*f_1_* 30, *f_1_*–*f_2_* 22, *h_1_*–*h_1_* 17, *h_1_*–*h_2_* 19. Prodorsal integument with weakly striate-spiculate ornamentation, hysterosomal ornamentation striate-spiculate.

Idiosomal venter (Figure 9 and Figure 13A). Coxisternal fields outlined with weakly striate-spiculate ornamentation, separated medially by striate-spiculate ornamentation of longitudinal course. Lengths of coxisternal setae: *1a* 12, *1b* 17, *1c* 10, *2b* 15, *3a* 11, *3b* 15, *3c* 16, *3d* 17, *4a* 11, *4b* 15, *4c* 17; distances: *1a*–*1a* 12, *1b*–*1b* 46, *1c*–*1c* 70, *2b*–*2b* 75, *3a*–*3a* 26, *3b*–*3b* 77, *3c*–*3c* 99, *3d*–*3d* 114, *4a*–*4a* 27, *4b*–*4b* 67, *4c*–*4c* 95. Genital region (Figure 10A and Figure 13C) with four pairs of aggenital setae: *ag_1_* 10 long, *ag_2_* 9 long, *ag_3_*_-*4*_ 6 long, and five genital setae: *g_1_* 13 long, *g_2_* 8 long, *g_3_*_-*4*_ 5 long, *g_3_* 6 long. Four pairs of *eu* setae, 6 long, on protuberances. Sternal setae (*1a*, *3a*, *4a*), genital, aggenital and *ps_3_* setae slightly expanded distally. Lengths of pseudanal setae: *ps_1_* 19, *ps_2_* 19, *ps_3_* 14.

Gnathosoma (Figure 10B–E and Figure 13D,E). Subcapitulum (Figure 10B) 51 long, 40 wide, slender, roughly triangular, with spiculate ornamentation. Subcapitular apodema visible under integument. Setae *sbc_1_* 10 long, *sbc_2_* 9 long, densely pilose, subequal. Chelicerae (Figure 10C) 60 long, 20 wide, with spiculate ornamentation, bearing small smooth dorsal seta *cha*; fixed digit with two pointed tips directed forward; movable digit sharp, clawlike. Palps (Figure 10D,E) 92 long, with spiculate ornamentation, spiculate-cuspidate on femorogenu. Palpal femorogenu 36 long. Palpal tibia 20 long, seta *l*″ 2/3 length of *l*′. Palpal tarsus 19 long, oval in lateral aspect, triangular in dorsoventral aspect. Setae *d*, *l*′, *l*″, *sl*″, *v* and *ba* pilose; setae *acm*, *p*′ and *p*″ smooth; rhagidial organ *ω* ellipsoid with proximal stock.

Legs (Figure 11, Figure 12 and Figure 13F,G). Lengths of legs: I 193, II 147, III 149, IV 190. Lengths of leg segments: I: Ts 68, Tb 32, G 29, F 71, Tr 28; II: Ts 43, Tb 25, G 23, F 50, Tr 23; III: Ts 46, Tb 26, G 22, TF 188, BF 35, Tr 25; IV: Ts 48, Tb 37, G 29, TF 25, BF 47, Tr 30. Integument with spiculate ornamentation, spiculate-cuspidate on basifemur III and from tibia to trochanter of leg IV. Leg setal formulae: I: 1–7+5–6(*σ*)–5(2*φ*, *κ*)–20(2*ω*, *ɛ*); II: 1–5+5–4(*σ*)–5(2*φ*)–13(3*ω*, *ɛ*); III: 1–4+4–3–4(*φ*)–12; IV: 0–3+3–3–5–11. Leg eupathidial setae: I: Tb: (*l*) Ts: all except (*v_3_*); II: Ts: *tc*′, (*p*); III: Tb: *d*, *v*″; Ts: (*p*); IV: G: *l*′; Tb: *d*, *l*′; Ts: (*p*).

Solenidia and famuli. Leg I. Genu with dorsomedial erect solenidion *σ*. Tibia with one anterior rhagidial organ *φ_1_* 3 long, associated with spiniform famulus *κ* and one medial, globular rhagidial organ *φ_2_* 1 long, in separate depressions. Tarsus with two T-shaped rhagidial organs: *ω_1_* 9 long, *ω_2_* 6 long, and stellate famulus *ɛ*, tandemly in confluent depression. Leg II. Genu with dorsomedial erect solenidion *σ*. Tibia with one short dorsodistal rhagidial organ *φ_1_* 3 long and dorsomedial globular rhagidial organ *φ_2_* 1 long. Tarsus with three T-shaped rhagidial organs *ω_1_* 7 long, *ω_2_*_,*3*_ 5 long, tandemly in confluent depression, subtended by spiniform famulus *ɛ*. Leg III. Tibia with globular rhagidial organ *φ* 1 long.

Differential diagnosis. *N. paulinae* resembles *N. parvus* by lack of seta on trochanter IV and short, globular proximal rhagidial organs on tibiae I and II. It differs from *N. parvus* in number of genital setae (five instead of six) and number of rhagidial organs on tarsus II (three instead of two).

Distribution. Hallett Peninsula, Antarctica [5].

Material examined. Holotype female (Bishop Museum, slide labeled “Bishop 7986”): Hallett Peninsula, Cape Hallett, about 1000 m southeast of Hallett Station on a talus slope, 72°20′ S 170°10′ E, loose soil in north shadow of rock, 25 December 1966, leg. E. Gless.

Remarks. Chaetotaxy of holotype differs significantly from that in original description by Gless [5]. The most apparent seems to be the discrepancy in genital chaetotaxy, i.e., six genital setae in original description and undoubtedly five in holotype female (Figure 10A and Figure 13C). On one hand, this can be attributable to misfortunate arrangement of the last pair of eugenital setae (*eu_4_*) which is everted outward the progenital chamber and supplants, (evidently lacking) last pair of genital setae (*g_6_*). On the other hand, the depicted body ventral side of a female and genital region of a male in the original paper (Figures 32 and 33 in [5]) clearly show six pairs of genital setae in both sexes. As no other specimens of *N. paulinae* are available for this study, it is impossible to decide if it is a both-sided anomaly in holotype or typical state of the species, and thus this character is excluded from the couplet No. 4 of the key.

Species formerly listed as, but not belonging to the genus *Neoprotereunetes* according to the newly proposed diagnosis:*Ereunetes lapidarius* Oudemans, 1906: a senior synonym of *Filieupodes filistellatus* Jesionowska, 2010 (Cocceupodidae).*Protereuntes maudae* Strandtmann, 1967: transferred herewith to the new genus *Antarcteupodes* gen. nov.*Protereunetes turgidus* Shiba, 1978: transferred by Khaustov (2017) to the genus *Echinoeupodes* Khaustov, 2017.*Protereunetes villosus* Shiba, 1978: probably belongs to the genus *Benoinyssus* Fain, 1958.*Protereunetes perforatus* Shiba, 1978: resembles *Caleupodes reticulatus* Baker, 1987, but it differs in body size and form of solenidia.


*Species Inquirenda*


*Protereunetes striatellus* (C.L. Koch, 1838): the species description is not sufficient to determine its generic affiliation and the type material most probably does not exist.

***Antarcteupodes* Laniecki** gen. nov.

Type species: *Protereunetes maudae* Strandtmann, 1967; monobasic.

Diagnosis. Sejugal furrow present. Idiosomal integument with lightly striate-spiculate ornamentation. Internal vertical setae (*v_1_*) inserted in bothridia, on well-delimited naso. Prodorsal trichobothria (*sc_1_*) filiform and pilose. Hysterosomal setae short, thin and setose. No hysterosomal trichobothria. Coxisternal setal formula: 3–1–3–2. Six pairs of genital setae in single row and none more lateral than others. Three pairs of pseudanal setae. Adanal setae absent. Four pairs of lyrifissures. Palpal setal formula: 0–2–3–8(*ω*). All legs shorter than body. Femora IV not enlarged. Leg integument with striate-spiculate ornamentation. Tibiae I and II each with one distal rhagidial organ and one proximal erect solenidion.

Description. Idiosomal dorsum. Sejugal furrow present. Integument with lightly striate-spiculate ornamentation. Prodorsum bearing four pairs of setae: *v_1_*, *v_2_*, *sc_1_* and *sc_2_*. Naso basally delimited from prodorsal shield and bearing short setae *v_1_* inserted in bothridia. Setae *sc_1_* trichobothrial, short, not reaching the posterior edge of naso. Remaining prodorsal setae short, setose, inserted in typical areolae and none of them trichobothrial. Hysterosoma bearing eight pairs of dorsal setae: *c_1_*, *c_2_*, *d_1_*, *e_1_*, *f_1_*, *f_2_*, *h_1_* and *h_2_*. All hysterosomal setae short, thin and setose. Three pairs of dorsal lyrifissures (*ia*, *im*, *ip*) present.

Idiosomal venter. Coxisternal fields integument with weakly striate-spiculate ornamentation. Coxisternal formula: 3–1–3–2; setae *3d* and *4c* not present. Small cavities near outer margin of coxae I-III present. Genital aperture postero-ventral, flanked by five pairs of aggenital setae (*ag_1_*_-*5*_). Genital valves bearing six pairs of genital setae (*g_1_*_-*6*_) of which anterior first is longer than second and second is longer than remaining ones. All setae *g* in single row and none more lateral than others. Internal genital structures consisting of two pairs of genital papillae and six pairs of eugenital setae (*eu_1_*_-*6*_) set on protuberances. Anal opening terminal, flanked by three pairs of pseudanal setae: *ps_1_*_,_ *_2_*, posteriorly and shorter *ps_3_*, anteriorly. No anal setae (*an*) on anal valves. One pair of ventral lyrifissures (*ih*) present.

Gnathosoma. Subcapitulum roughly triangular, squat bearing four pairs of setae: two pairs of setose subcapitular setae (*sbc_1_*_,*2*_) and two pairs of minute smooth adoral setae (*or_1_*_,_ *_2_*). Setae *sbc_1_* thinner and shorter than *sbc_2_*, both located along the base of each lateral lip, at antiaxial and paraxial end of subcapitular apodema, respectively. Setae *or_1_* and *or_2_* closely clustered at the tip of each lateral lip and hard to discern. Apex of labrum acuminate. Chelicerae thick, bearing long, nude dorsal seta *cha*. Palps four-segmented with weakly barbed supracoxal seta *ep*. Palpal setal formula: 0–2–3–8(*ω*). Palparsus laterally flattened, bearing eight setae: *d*, *l*′, *l*″, *acm*, *p*′, *p*″, *v*, *ba* and small rhagidial organ *ω*; seta *sl*″ not present. Cheliceral and palpal ornamentation spiculate (spiculate-cuspidate on palpal femorogenu).

Legs. Legs I and IV longer than II and III, but all shorter than body. Femora of I and II leg undivided. Femora III and IV divided. All apoteles consist of pad-like empodium with dense setulae arranged in bands on lateral margins and pair of hooked claws with short outgrows on its ventral surface. Integument with striate-spiculate ornamentation. All setae setose except weakly barbed supracoxal seta *el*. Solenidia and famuli. Leg I. Genu with one dorsomedial erect solenidion *σ*. Tibia with one anterior complex of short ellipsoid rhagidial organ *φ* and weakly furcate famulus *κ*, and one proximal short erect solenidion. Tarsus with two L-shaped rhagidial organs (*ω*) and one small weakly stellate famulus *ɛ*, tandemly in separated depressions. Leg II. Genu without solenidion. Tibia with one ellipsoid rhagidial organ *φ* and one proximal erect solenidion. Tarsus with three L-shaped rhagidial organs and with weakly furcate famulus *ɛ*, arranged alternately, i.e., anterior and posterior rhagidial organs situated antiaxially, whereas medial one—paraxially, each in separated depression. Leg III. Genu without solenidion. Tibia with proximal erect solenidion *φ*. Tarsus without rhagidial organs. Leg IV. Genu without solenidion. Tibia with proximal erect solenidion *φ*. Tarsus without rhagidial organs.

Differential diagnosis. The new genus is similar to *Pseudoeupodes* Khaustov, 2014 because of legs shorter than body, femora IV not enlarged, short dorsal setae, and number and location of genital setae. It differs from *Pseudoeupodes* by internal vertical setae located in bothridia (in common areolae in *Pseudoeupodes*), coxisternal formula: 3–1–3–2 (3–1–4–2 in *Pseudoeupodes*) and three pairs of pseudanal setae (two in *Pseudoeupodes*). It resembles also *Neoprotereunetes* Fain et Camerik, 1994 in having short dorsal setae, same number and location of genital setae, all legs shorter than body, and not enlarged femora IV. It differs from *Neoprotereunetes* in internal vertical setae located in bothridia (in common areolae in *Neoprotereunetes*), coxisternal formula: 3–1–3–2 (3–1–4–3 in *Neoprotereunetes*) and in presence of one rhagidial organ and one erect solenidion on both tibiae I and II (two rhagidial organs in *Neoprotereunetes*).

***Antarcteupodes maudae* (Strandtmann, 1967)** comb. nov. (Figure 14, Figure 15, Figure 16, Figure 17, Figure 18, Figure 19 and Figure 20)

*Protereunetes maudae* [22,27,29]

Redescription. Holotype female. Idiosoma flattened and ruptured along its right margin, 360 long, ca. 220 wide.

Idiosomal dorsum (Figure 14 and Figure 20A). Prodorsal shield 74 long, 100 wide, triangular. Prodorsal integument with weakly striate-spiculate ornamentation, but course of striae hard to retrace. Naso (Figure 20B) 15 long, 28 wide, rounded. A pair of canals, probably representing tracheae (*tr*?), extending from anterior end of idiosoma to posterior corners of prodorsum (Figure 14). Lengths of prodorsal setae: *v_1_* 18, *v_2_* 14, *sc_1_* ca. 40, *sc_2_* 18; distances: *v_1_*–*v_1_* 8, *v_2_*–*v_2_* 60, *sc_1_*–*sc_1_* 37, *sc_2_*–*sc_2_* 92. Hysterosoma roughly rectangular, slightly rounded caudally. Lengths of hysterosomal setae: *c_1_* 14, *c_2_* 24, *d_1_* 14, *e_1_* 14, *f_1_* 19, *f_2_* 20, *h_1_* 24, *h_2_* 23; distances: *c_1_*–*c_1_* 50, *c_1_*–*c_2_* 78, *d_1_*–*d_1_* 58, *e_1_*–*e_1_* 63, *f_1_*–*f_1_* 52, *f_1_*–*f_2_* 33, *h_1_*–*h_1_* 22, *h_1_*–*h_2_* 27. Prodorsal and hysterosomal integument with lightly striate-spiculate ornamentation.

Idiosomal venter (Figure 15 and Figure 20C). Coxisternal fields outlined, with weakly striate-spiculate ornamentation, separated medially by striate-spiculate ornamentation of longitudinal course. Lengths of coxisternal setae: *1a* 14, *1b* 16, *1c* 10, *2b* 13, *3a* 12, *3b* 15, *3c* 15, *4a* 10, *4b* 15; distances: *1a*–*1a* 27, *1b*–*1b* 69, *1c*–*1c* 97, *2b*–*2b* 82, *3a*–*3a* 39, *3b*–*3b* 92, *3c*–*3c* 122, *4a*–*4a* 42, *4b*–*4b* 104. Coxal cavities weakly defined, half-open. Genital region (Figure 16A) with five pairs of aggenital setae: *ag_1_* 10 long, *ag_2_* 9 long, *ag_3_*_-*4*_ 8 long, *ag_5_* 7 long, and six pairs of genital setae: *g_1_* 10 long, *g_2_* 9 long, *g_3_*_-*6*_ 7 long. Six pairs of eugenital setae, ca. 10 long, on protuberances (Figure 16B). Genital, aggenital and *ps*_3_ setae slightly expanded distally. Lengths of pseudanal setae: *ps_1_* 24, *ps_2_* 19, *ps_3_* 9.

Gnathosoma (Figure 16C, Figure 17A–C and Figure 20D,E). Subcapitulum (Figure 17A) 47 long, 50 wide. Border between lateral lips and subcapitular base visible under integument. Setae *sbc_2_* 7 long, setose, thicker than sparsely setose *sbc_1_* 5 long. Chelicerae (Figure 16C) 70 long, 29 wide, with small smooth dorsal seta *cha*; fixed digit with two pointed tips directed forward; movable digit sharp, clawlike. Palps (Figure 17B,C and Figure 20D,E) 118 long, with striate-spiculate ornamentation, spiculate-cuspidate on femorogenu. Palpal femorogenu 50 long. Palpal tibia 25 long, seta *l*″ 2/3 length of two times thicker *l*′. Palpal tarsus 37 long, ellipsoid in dorsoventral aspect. Setae *d*, *l*′, *l*″, *v* and *ba* pilose; setae *acm*, *p*′ and *p*″ smooth; rhagidial organ *ω* minute, protruding on both tarsi in dorsoventral view. Subcapitular, cheliceral and palpal integument with striate-spiculate ornamentation (spiculate-cuspidate on palpal femorogenu).

Legs (Figure 18, Figure 19 and Figure 20F,G). Lengths of legs: I 247, II 197, III 200, IV 255. Lengths of leg segments: I: Ts 64, Tb 45, G 35, F 88, Tr 35; II: Ts 53, Tb 34, G 30, F 68, Tr 28; III: Ts 53, Tb 37, G 27, TF 30, BF 38, Tr 32; IV: Ts 63, Tb 43, G 35, TF 35, BF 60, Tr 43. Integument with striate-spiculate ornamentation, spiculate-cuspidate on basifemur of leg III and from genu to basifemur of leg IV. Leg setal formulae: I: 1–3+5–4(*σ*)–5(2*φ*, *κ*)–17(2*ω*, *ɛ*); II: 1–2+5–4–5(2*φ*)–11(3*ω*, *ɛ*); III: 1–2+3–3–3(*φ*)–11; IV: 1–2+3–3–4(*φ*)–9. Eupathidial setae: I: Tb: all; Ts: all and *v_2_*″?; II: Tb: *d* (only on left leg), *l*″, *v*″; Ts: all; III: Tb: *d*?, *v*′?; Ts: all and (*tc*)? (*it*)?; IV: Tb: *d*?; Ts: all and *tc*?, (*it*)?. Solenidia and famuli as in generic description. Lengths of rhagidial organs: leg I: *φ_1_* 4, *ω_1_* 8, *ω_2_* 5; leg II: *φ_1_* 4, *ω_1_* 8, *ω_2_* 8, *ω_3_* 8.

Distribution. Victoria Land, Antarctica [29].

Material examined. Holotype female (Bishop Museum, slide labeled “BBM 7056”): Shackleton Glacier area, north of Garden Spur, east side of Massam Glacier, 457 m elevation, 84°33′ S 174°40′ E, 15 December 1964, leg. J. Shoup.

Remarks. The species is characterized by the unique combination of character states, not present in any hitherto described eupodid genus, including the most reduced coxisternal and leg chaetotaxy among the family Eupodidae, and sufficient to represent a separate genus.

Family: Cocceupodidae Jesionowska, 2010

*Filieupodes* Jesionowska, 2010

Type species: *Filieupodes filiformis* Jesionowska, 2010 by original designation.

***Filieupodes lapidarius* (Oudemans, 1906)** comb. nov. (Figure 21, Figure 22, Figure 23, Figure 24, Figure 25 and Figure 26)

*Ereunetes lapidarius* [34]

*Ereynetes lapidarius* [35,36]

*Micrereunetes (Protereunetes) lapidarius* [6]

*Protereunetes lapidarius* [7,8,11]

*Neoprotereunetes lapidarius* [16,22]

*Filieupodes filistellatus* Jesionowska, 2010 syn. nov.

Diagnosis. Naso well delimited. Dorsal hysterosomal setae short. Tarsus I with two parallel rhagidial organs in separate depressions, of which proximal one posterolaterad of distal one. Stellate famulus well removed proximo-laterally from proximal rhagidial organ. Tarsus II with three parallel rhagidial organs in separate depressions, of which medial one posterolaterad of proximal and distal ones. Spiniform famulus well removed laterally from proximal rhagidial organ.

Redescription. Holotype female. Idiosoma 330 long, 200 wide.

Idiosomal dorsum (Figure 21 and Figure 26A). Prodorsal shield (Figure 26B) 74 long, 100 wide, triangular. Prodorsal integument with weakly striate-spiculate ornamentation, but course of striae hard to retrace. Naso 15 long, 28 wide, rounded. Lengths of prodorsal setae: *v_1_* 34, *v_2_* 22, *sc_1_* ca. 50, *sc_2_* 25. Hysterosoma oval. Hysterosomal integument with striate-spiculate ornamentation. Lengths of hysterosomal setae: *c_1_* 24, *c_2_* 40, *d_1_* 27, *e_1_* 30, *f_1_* 40, *f_2_* 33, *h_1_* 41, *h_2_* 27.

Idiosomal venter (Figure 22). Coxisternal fields poorly outlined, with weakly striate-spiculate ornamentation, separated medially by striate-spiculate ornamentation of longitudinal course. Lengths of coxisternal setae: *1a* 18, *1b* 20, *1c* 13, *2b* 26, *3a* 18, *3b* 18, *3c* 20, *3d* 18, *4a* 13, *4b* 16, *4c* 15. Genital region (Figure 23A and Figure 26C) with four pairs of aggenital setae, *ag* 9 long, *g* 10 long and six pairs of genital setae, all ca. 10 long. Two pairs of genital papillae and five pairs of eugenital setae, 8 long, on protuberances. Lengths of pseudanal setae: *ps_1_* 32, *ps_3_* 15. Lyrifissures *ih* not visible.

Gnathosoma (Figure 23B–D and Figure 26D,E). Subcapitulum (Figure 23B) 52 long, 46 wide roughly triangular, with striate-spiculate ornamentation. Subcapitular apodema visible under integument. Setae *sbc_2_*, 9 long, subequal to *sbc_1_*, 12 long, both pilose. Chelicerae (Figure 23C) 60 long, with spiculate ornamentation, bearing pilose dorsal seta *cha*; fixed digit with blunt tip; movable digit sharp, clawlike. Palps (Figure 23D and Figure 26D,E) with spiculate ornamentation. Palpal femorogenu 29 long. Palpal tibia 33 long, setae *l*″ and *l*′ subequal in length. Palpal tarsus 20 long, ellipsoid in dorsoventral aspect. All setae except *acm* smooth; rhagidial organ *ω* small, protruding on both tarsi in dorsoventral view. Subcapitular, cheliceral and palpal integument with spiculate ornamentation (spiculate-cuspidate on palpal femorogenu).

Legs (Figure 24, Figure 25 and Figure 26F,G). Lengths of legs: I 315, II 198, III 221, IV 255. Lengths of leg segments: I: Ts 77, Tb 64, G 51, F 112, Tr 27; II: Ts 58, Tb 37, G 24, F 74, Tr 20; III: Ts 62, Tb 37, G 32, TF 29, BF 41, Tr 20; IV: Ts 72, Tb 46, G 51, TF 23, BF 64, Tr 23. Integument with spiculate ornamentation. Leg setal formulae: I: 1–6+5–8–13(2*φ*)–21(2*ω*, *ɛ*); II: 1–5+5–4–5(2*φ*)–12(3*ω*, *ɛ*); III: 1–4–4–4–5–12; IV: 1–3–3–4–5–12. Eupathidial setae: I: G: (*l*); Tb: all except (*l_1_*_-*2*_); Ts: all; II: Tb: *d*, *v*′; Ts: all except *ft*′; III: G: *l*′; Tb: *d*, *l*′; Ts: all; IV: BF *d*; G: *l*″; Tb: *d*, *l*′; Ts: all.

Solenidia and famuli. Leg I. Tarsus with two parallel T-shaped rhagidial organs in confluent depression and one stellate famulus *ɛ* well moved antiaxially to the lateral side. Posterior, dorsolateral rhagidial organ (*ω_1_*) reaching half of the length of anterior, dorsal one (*ω_2_*). Tibia with one distal (*φ_1_*) and one medial rhagidial organ *φ_2_*, both T-shaped, tandemly in separated depressions. Leg II. Tarsus with three parallel T-shaped rhagidial organs in separated depressions, of which the smallest anterior one (*ω_3_*) oblique antiaxially and flanked by two bigger posterior ones (*ω_1_* and *ω_2_*). Spiniform famulus *ɛ* not visible. Tibia with two T-shaped rhagidial organs (anterior *φ_1_* and medial *φ_2_*), in separated depressions. Leg III and IV without solenidia.

Differential diagnosis. *F. lapidarius* is similar to *F. shepardi* Strandtmann, 1971 because of naso delimited dorsally and the same number of aggenital and genital setae. It differs from *F. shepardi* in short dorsal hysterosomal setae (long in *F. shepardi*) and parallel arrangement of rhagidial organs on tarsi I and II (tandem in *F. shepardi*).

Distribution. Arnhem, Netherlands [34]; nature reserve of halophitic vegetation, Ciechocinek near Toruń, Kujawsko-Pomorskie District; “Zielona Góra” nature reserve, vicinity of Częstochowa, Śląskie District (both [4]); Morasko Campus, Poznań, Wielkopolskie District [37], all latter localities in Poland.

Material examined. Holotype female (Naturalis Biodiversity Centre, slide labeled “RMNH.ACA.P 5507”): Netherlands, Arnhem, under stones, 1903, leg. Dammermann; four females and two males: Poland, Wielkopolskie district, Poznań, Morasko University Campus, 52°27′58″ N 16°55′21″ E, Fresh meadow with often reaped *Arrhenatheretum elatioris*, 19 February 2019, leg. R. Laniecki.

Remarks. The original description [34], as well as subsequent redescriptions [35,36], lacks some valid diagnostic characters and thus the species is redescribed herewith. Despite high similarity of the holotype of *Filieupodes lapidarius* and specimens previously identified as *F. filistellatus* collected in Poland, some differences can be observed. In the original description of *F. filistellatus* proximal rhagidial organs are present on all tibiae, whereas they were not found on tibiae III and IV in holotype of *F. lapidarius*. This, however, could be a result of age and condition of the original material (117 years). Because of this, some structures (e.g., lyrifissures, supracoxal setae) are not visible. Moreover, due to the position of the specimen on the slide, some structures could not be entirely distinguished, e.g., some coxisternal and aggenital setae. Those, however, which are visible, fit the setal patterns of *F. filistellatus*.

The type material of *F. filistellatus* is lost (courtesy of Prof. Andrzej J. Zawal, former superior of Dr. Katarzyna Jesionowska), and thus only newly collected material along with the original description by Jesionowska [4] were used to compare the species with holotype of *N. lapidarius*.

### 3.2. Key to the Species of Neoprotereunetes (Adults)

1. Five pairs of genital setae, tarsus I with 21 setae, femur I with 13 setae, tibia II with five setae, genu III and IV each with four setae, arctic species .................................................... ......... *boerneri* species group ........................................................................ *boerneri* (Thor, 1934)

– Five or six pairs of genital setae, tarsus I with 20 setae, femur I with 12 setae, tibia II with four setae, genu III and IV each with three setae, Antarctic or sub-Antarctic species ......... *minutus* species group ......................................................................................................... 2

2. Tibial proximal rhagidial organs long, at most four times shorter than its segment, famulus *ε* on tarsus II absent ....................................................................................................... 3

– Tibial proximal rhagidial organs short, at least seven times shorter than its segment, famulus *ε* on tarsus II present ..................................................................................................... 4

3. Tibia III with proximal rhagidial organs, two anterior rhagidial organs (*ω_2_*_,_ *_3_*) on tarsus II parallel and arranged side by side, both tarsal rhagidial organs in confluent depressions.....................................................................................*minutus* (Strandtmann, 1967)

– Tibia III without proximal rhagidial organ, two anterior rhagidial organs (*ω_2_*_,_ *_3_*) on tarsus II parallel, but *ω*_3_ displaced anteriorly in relation to *ω*_2_, both tarsal rhagidial organs in separated depressions ................................................ *crozeti* (Strandtmann et Davies, 1972)

4. Tarsus II with three rhagidial organs of unequal size, tibia III with short ellipsoid rhagidial organ, trochanter IV with one seta ............ *exiguus* (Booth, Edward et Usher, 1985)

– Tarsus II with three rhagidial organs of equal size, tibia III with small spherical rhagidial organ, trochanter IV without setae .......................................... *paulinae* (Gless, 1972)

– Tarsus II with two rhagidial organs, tibia III without rhagidial organ, trochanter IV without setae .................................................................. *parvus* (Booth, Edward et Usher, 1985)

## 4. Discussion

The family Eupodidae is composed mostly of monotypic genera, e.g., *Claveupodes*, *Caleupodes*, *Aethosolenia*. Two non-monotypic genera, i.e., *Pseudopenthaleus* and *Echinoeupodes*, have two species each, but in both cases, only one of them is accurately described. The remaining two non-monotypic genera, i.e., *Eupodes* and *Benoinyssus* are highly heterogenous. It is, therefore, hard to establish diagnostic characters at the generic level. We decided not to base generic diagnoses on leg setal patterns until more data on intra-generic variability in this respect will be collected. Body dimensions and shape are also excluded from diagnoses as these characters are contingent on age and condition of an individual as well as specimen treatment and preparation technique and may even change with an age of the slide.

In the present study, six species were classified within the genus *Neoprotereunetes*. Though the Arctic species differ slightly from Antarctic and sub-Antarctic congeners (mostly in genital and leg chaetotaxy), we decided not to divide them into separate genera or subgenera until the intrageneric variability in eupodid genera is better understood. However, to express these differences, two species groups were proposed: one, *boerneri*, containing *N. boerneri*, and another one, *minutus,* containing *N. crozeti*, *N. exiguus*, *N. minutus*, *N. parvus* and *N. paulinae*, based on type or newly collected material as well as original descriptions and redescriptions. Additionally, seven species that were described in or transferred to *Protereunetes* are not included in *Neoprotereunetes*. The first one was originally described by Oudemans [34] as *Ereunetes lapidarius* (Ereynetidae). In [35], the description of this species, with the (then) corrected generic name (*Ereynetes*), was extended and supplied with pictures by the same author. Next Oudemans [36] moved *E. lapidarius* to the family Eupodidae, without reference to its generic rank. Subsequently, Thor [7] transferred *E. lapidarius* to the genus *Protereunetes*. Finally, this species was designated as a type species of *Neoprotereunetes* Fain et Camerik, 1994. However, after the present examination of the holotype, it turned out that *Neoprotereunetes lapidarius* is the senior synonym of *Filieupodes filistellatus* Jesionowska, 2010 (Cocceupodidae). The second one, *P. maudae*, described as a congener of *N. minutus* by Strandtmann [29], is designated as a type species of the new genus *Antarcteupodes* on the basis of its unique combination of character states, not present in any hitherto described eupodid genus, including the most reduced coxisternal and leg chaetotaxy among the family Eupodidae. The third one, *Protereunetes turgidus* Shiba, 1978, was transferred by Khaustov [26] to the genus *Echinoeupodes* Khaustov, 2017. The fourth, *Protereunetes villosus* Shiba, 1978, possesses long and slender setae *f*_1_ (presumably trichobothrial) and characteristic solenidiotaxy of tarsi I and II (each with two rhagidial organs, of which distal one is much smaller than proximal one), suggesting its affiliation to *Benoinyssus* Fain, 1958. The fifth species, *Protereunetes perforatus* Shiba, 1978, resembles *Caleupodes reticulatus* Baker, 1987 with respect to its reticulated body ornamentation and almost smooth setae. These characters are extremely rare in the family Eupodidae and might suggest a close relationship between these two species. Even if so, *P. perforatus* slightly differs from *C. reticulatus* in the solenidiotaxy of tarsus II (three rhagidial organs, instead of two) and the tibiae (one rhagidial organ, instead of two), and also in terms of its much larger body. The last species, *Protereunetes striatellus* (C.L. Koch, 1838) was originally described in *Eupodes* and then transferred by Thor and Willmann [8] to *Protereunetes*. As the description of this species is insufficient to determine its generic affiliation, and the type material probably does not exist, it is considered a *species inquirenda*. To confirm the above proposals, the type material should be examined, if (or when) available.

In reply to the transfer of *Protereuntes* (junior synonym of *Ereynetes*) back to Ereynetidae by Fain [9], Strandtmann [10] moved one of his species (*P. minutus*) to the genus *Eupodes* without reference to the second one (*P. maudae*). Although in subsequent papers Strandtmann was still using the name *Protereunetes* in relation to eupodoid mites, the usage of *Eupodes* sensu [10] was widely accepted by other authors [13,14,15]. However, in our opinion, the six species assigned herein to *Neoprotereunetes* possess a set of characters sufficient to constitute a separate genus. They have short and plumose dorsal body setae (long and lightly plumose in *Eupodes*); normal setae *f*_1_ (sometimes trichobothrial in *Eupodes*); all legs shorter than body (legs I and II longer than body in *Eupodes*); femur IV slender (usually swollen in *Eupodes*) short and plumose leg setae (long and pilose in *Eupodes*); two or three L-shaped or T-shaped rhagidial organs on tarsi I and II (always two L-shaped rhagidial organs in *Eupodes*); two rhagidial organs on tibiae I and II (one rhagidial organ and one erect solenidion in *Eupodes*). *Eupodes* is still a highly heterogeneous taxon demanding a major revision. Nevertheless, the abovementioned characters enable the separation of *Neoprotereunetes* from *Eupodes*.

According to the Principle of Priority [38], the synonymy of original type species of *Neoprotereunetes*, namely *Ereunetes lapidarius* Oudemans, 1906 with *Filieupodes filistellatus* Jesionowska, 2010 implies that *Neoprotereunetes* is the valid genus-level name and should replace *Filieupodes* as its senior synonym.

This, however, does not resolve the problem of the lack of a replacement for the genus-level name *Protereunetes*—the primary aim of creating *Neoprotereunetes* by Fain and Camerik [16]. As the descriptions, redescriptions and original figures of *E. lapidarius* [33,34,35] do not imply that this species belongs to the family Cocceupodidae, only the present examination of the type could demonstrate that. As the type species fixation of the genus *Neoprotereunetes* was based on a misidentification (even at the time of its inception it did not meet its own diagnosis) for the sake of nomenclatural stability, we suggest retaining the name *Filieupodes* for the genus in the family Cocceupodidae (in line with the original proposal by Jesionowska [4]) and *Neoprotereunetes* for the genus in the family Eupodidae (as used by Khaustov [17]).

Thus, because the designation of new type species for *Neoprotereunetes* becomes necessary, we propose establishing *Protereunetes boerneri* Thor, 1934 (the oldest known species after *E. lapidarius* listed by Thor and Willamnn [8]) as the type species of the newly diagnosed genus.

Such practices are justified and encouraged by ICZN [38], as expressed in its initial chapter “Introduction. Development of underlying principles” (p. 14) by the following statement: “Also when individual zoologists discover that the type species had been misidentified when a genus or subgenus was established, they are given the power to fix as the type species either the species actually nominated by the original author or the nominal species in conformity with the name in use”.

The representatives of *Neoprotereunetes* thus far have been found only in the high latitudes of either hemisphere. The *boerneri* species group is restricted to the Arctic (Svalbard, Severnaya Zemlya, Arctic Alaska) and sub-Arctic (sub-Arctic Alaska) locations, whereas the *minutus* species group is restricted to the Antarctic (e.g., Antarctic Peninsula, South Orkney Islands, South Shetland Islands) and sub-Antarctic (Crozet Islands, Prince Edward Islands) locations. Additionally, *N. minutus* has also been recorded in Dunedin, New Zealand. Apart from the latter, all the locations are characterized by harsh climate and low yearly temperatures that seem to be favorable to eupodoid mites. Among terrestrial Prostigmata, Eupodoidea dominate in the Antarctic (36 species described) are one of the dominating groups in the Arctic.

## 5. Conclusions

Establishing *Neoprotereunetes* as a replacement for *Protereunetes* constitutes an important step in dividing the large and highly heterogeneous eupodid genus *Eupodes* and contributes to increasing the stability within Eupodidae. Even though *Neoprotereunetes* displays no unique characters specific only to this genus, it can be easily defined by a combination of characters. This might be one of the reasons that it remained so poorly defined for such a long time.

## Figures and Tables

**Figure 1 animals-13-02213-f001:**
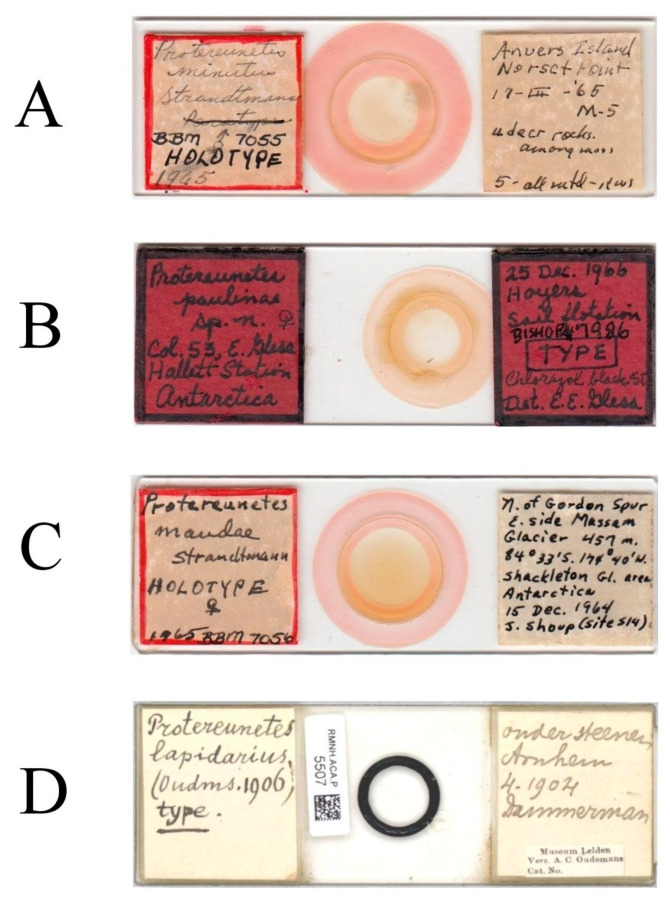
Microscopic slides. (**A**) *Neoprotereunetes minutus* (Strandtmann, 1967), holotype male; (**B**) *Neoprotereunetes paulinae* (Gless, 1972), holotype female; (**C**) *Antarcteupodes maudae* (Strandtmann, 1967), holotype female; (**D**) *Filieupodes lapidarius* (Oudemans, 1906), holotype female.

**Figure 2 animals-13-02213-f002:**
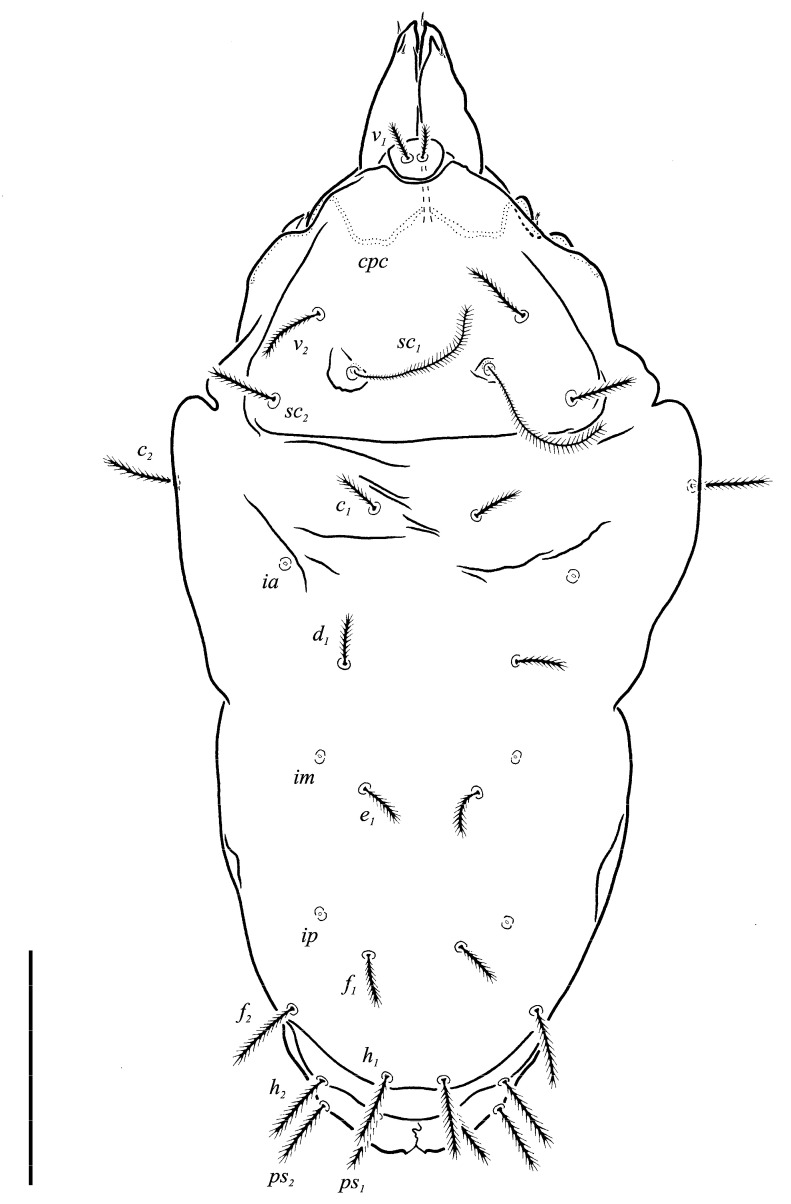
*Neoprotereunetes boerneri* (Thor, 1934), female. Body, dorsal view. Scale bar: 50 μm.

**Figure 3 animals-13-02213-f003:**
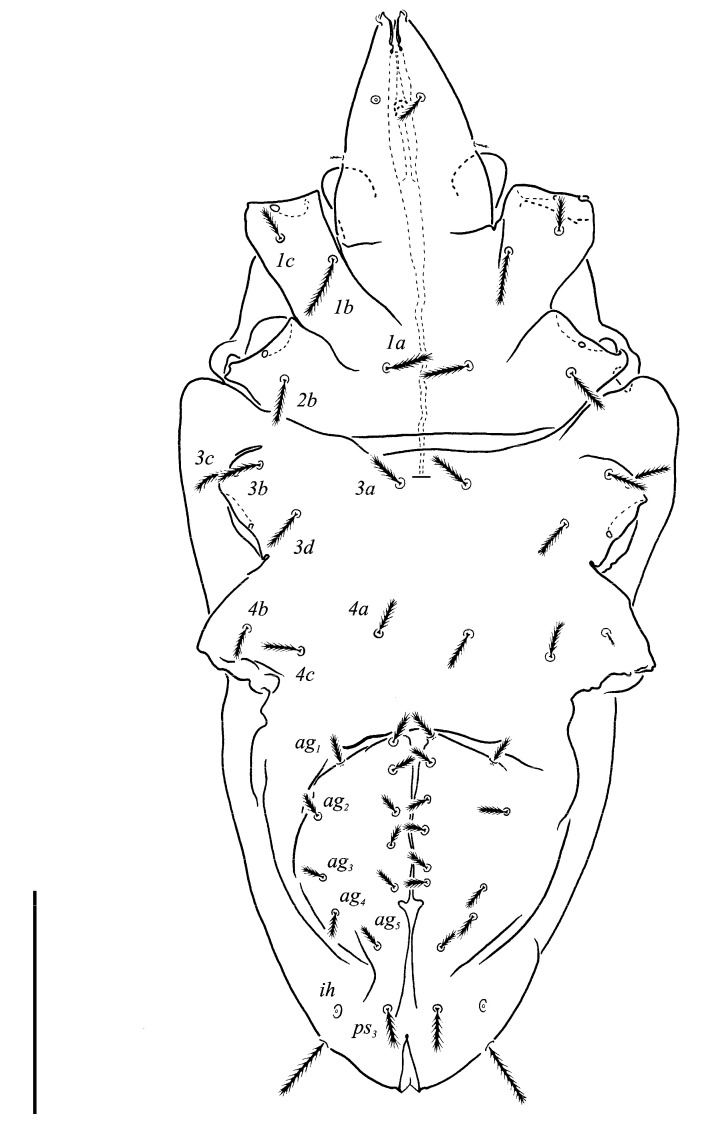
*Neoprotereunetes boerneri* (Thor, 1934), female. Body, ventral view. Scale bar: 50 μm.

**Figure 4 animals-13-02213-f004:**
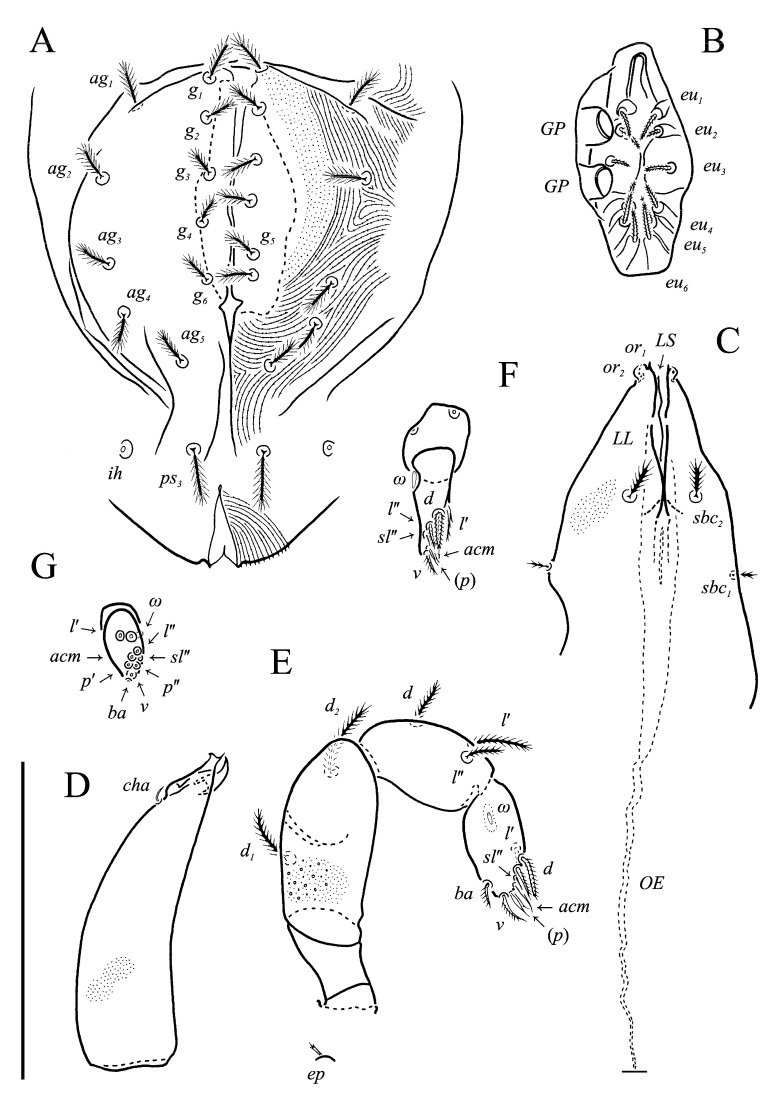
*Neoprotereunetes boerneri* (Thor, 1934), female. (**A**) Genital and anal region; (**B**) progenital chamber; (**C**) subcapitulum, ventral view; (**D**) right chelicera, lateral view; (**E**) left palp, lateral view; (**F**) tarsus of left palp, dorsal view; (**G**) tarsus of right palp, apical view. Scale bar: 50 μm.

**Figure 5 animals-13-02213-f005:**
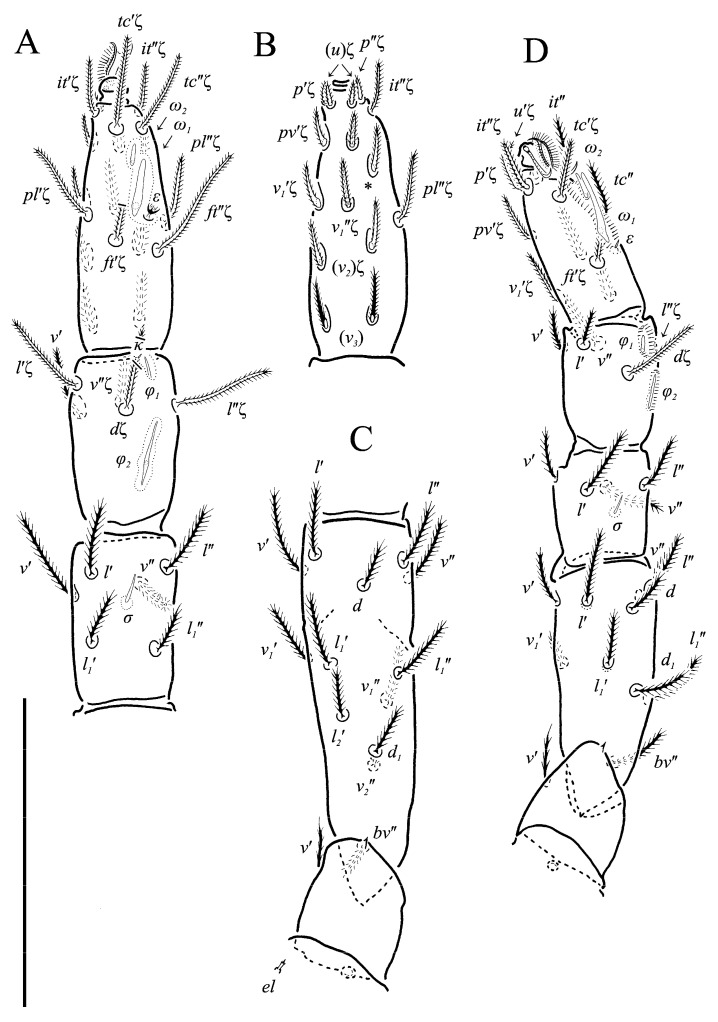
*Neoprotereunetes boerneri* (Thor, 1934), female. (**A**) Tarsus, tibia and genu of right leg I, dorsal view; (**B**) tarsus of right leg I, ventral view (apotele omitted); (**C**) femur and trochanter of right leg I, dorsal view; (**D**) right leg II, dorsolateral view. Asterisk denotes unpaired tarsal seta. Scale bar: 50 μm.

**Figure 6 animals-13-02213-f006:**
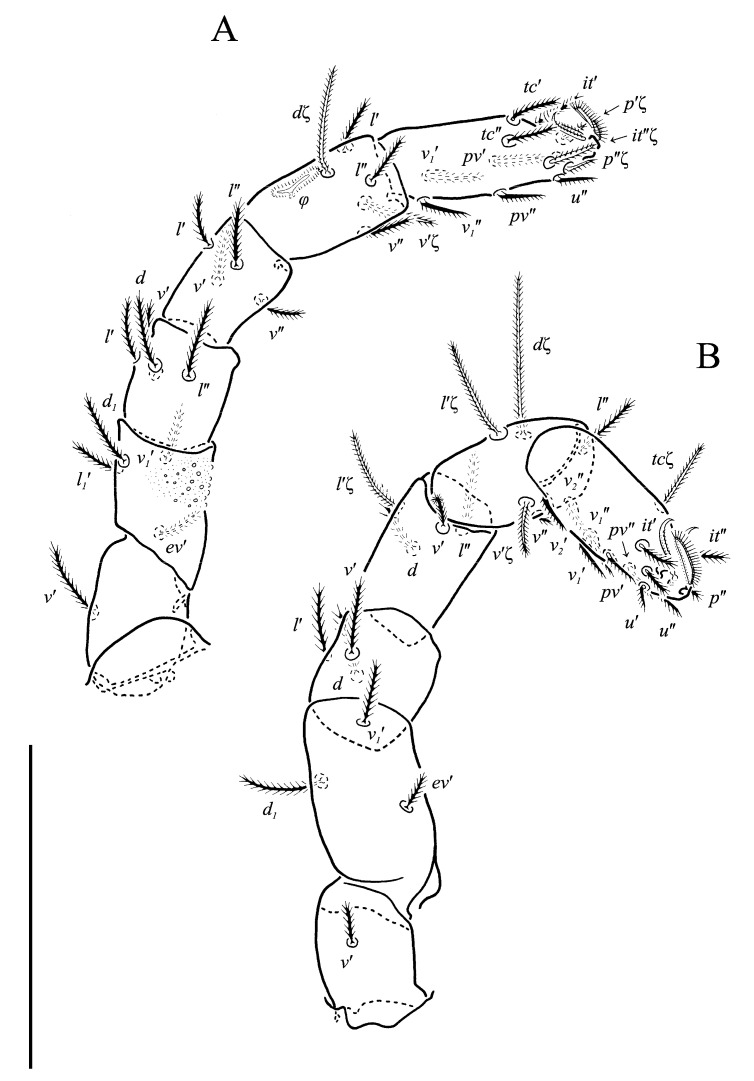
*Neoprotereunetes boerneri* (Thor, 1934), female. (**A**) Right leg III, dorsolateral view; (**B**) right leg IV, lateral view. Scale bar: 50 μm.

**Figure 7 animals-13-02213-f007:**
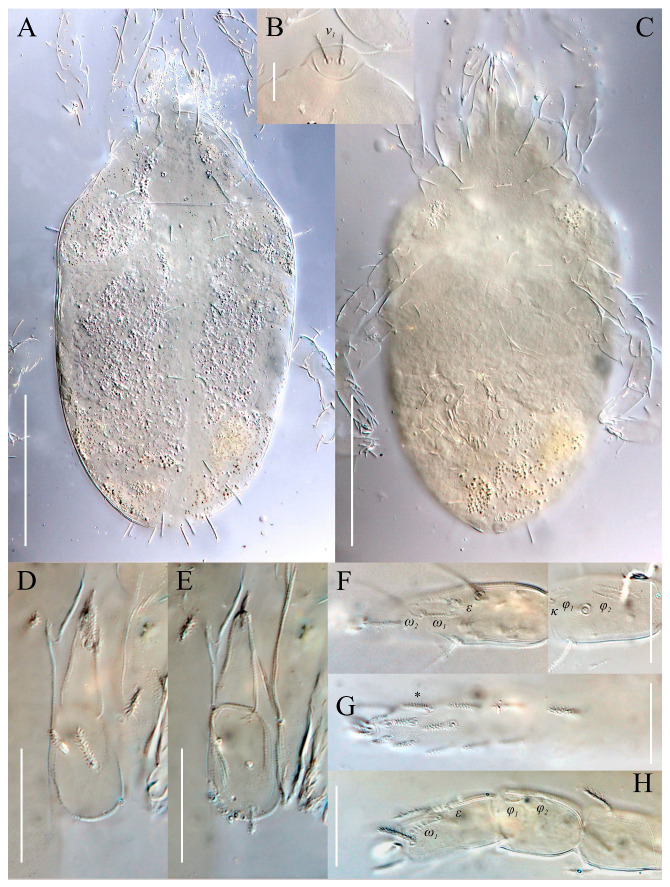
*Neoprotereunetes boerneri* (Thor, 1934), female. (**A**) Body, dorsal view; (**B**) naso; (**C**) body, ventral view; (**D**) tarsus and tibia of left palp, dorsal view; (**E**) tarsus and tibia of left palp, ventral view; (**F**) tarsus and tibia of right leg I, dorsal view; (**G**) tarsus of right leg I, ventral view; H—tarsus and tibia of right leg II, dorsolateral view. Asterisk denotes unpaired tarsal seta. Scale bar: (**A**,**C**) 100 μm; (**B**) 10 μm; (**D**–**H**) 20 μm.

**Figure 8 animals-13-02213-f008:**
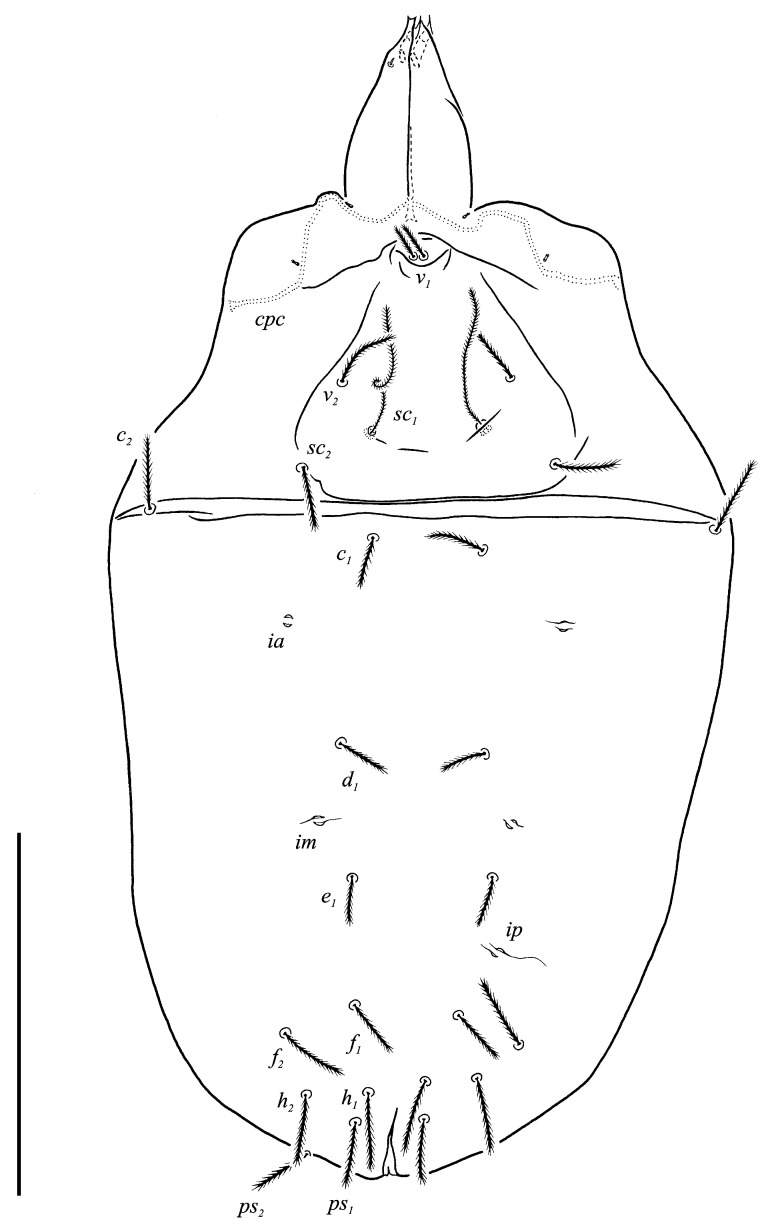
Neoprotereunetes paulinae (Gless, 1972), holotype female. Body, dorsal view. Scale bar: 100 μm.

**Figure 9 animals-13-02213-f009:**
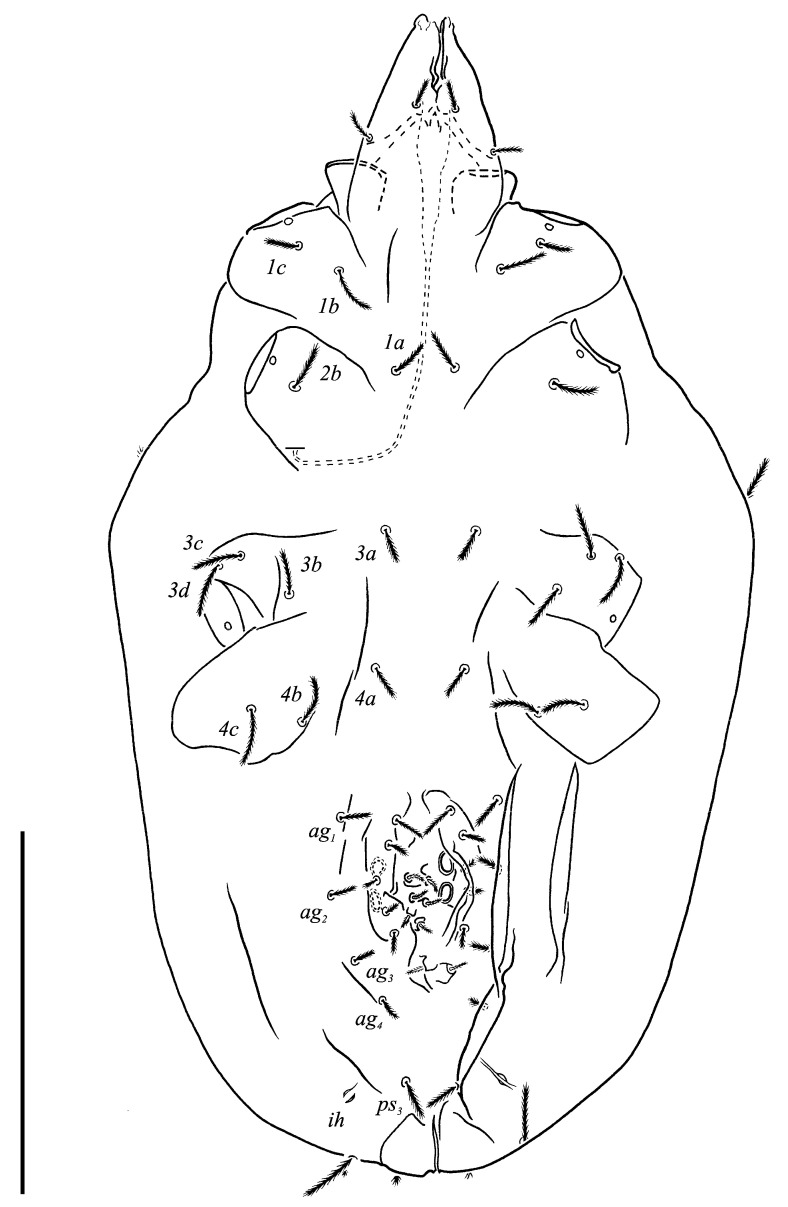
*Neoprotereunetes paulinae* (Gless, 1972), holotype female. Body, ventral view. Scale bar: 100 μm.

**Figure 10 animals-13-02213-f010:**
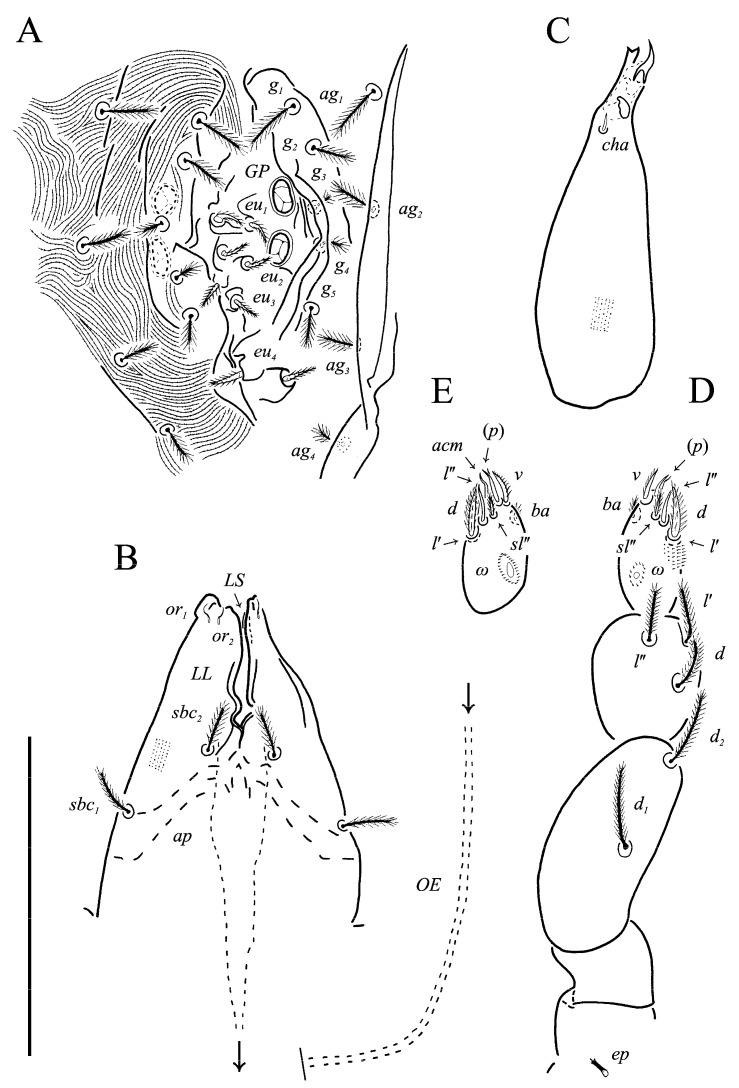
*Neoprotereunetes paulinae* (Gless, 1972), holotype female. (**A**) genital region; (**B**) subcapitulum, ventral view; (**C**) left chelicera, lateral view; (**D**) left palp, dorsolateral view; (**E**) tarsus of right palp, dorsolateral view. Scale bar 50 μm.

**Figure 11 animals-13-02213-f011:**
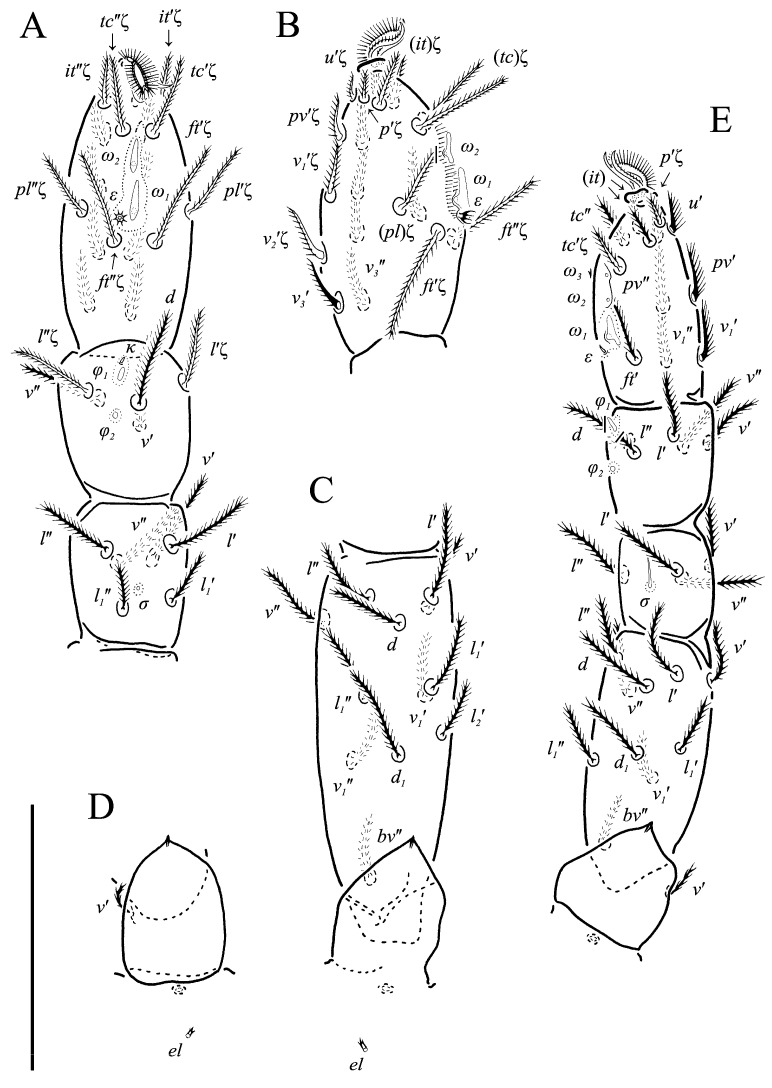
*Neoprotereunetes paulinae* (Gless, 1972), holotype female. (**A**) Tarsus, tibia and genu of left leg I, dorsal view; (**B**) tarsus of left leg I, lateral view; (**C**) femur and trochanter of left leg I, dorsal view; (**D**) trochanter of right leg I, dorsal view; (**E**) left leg II, dorsolateral view. Scale bar: 50 μm.

**Figure 12 animals-13-02213-f012:**
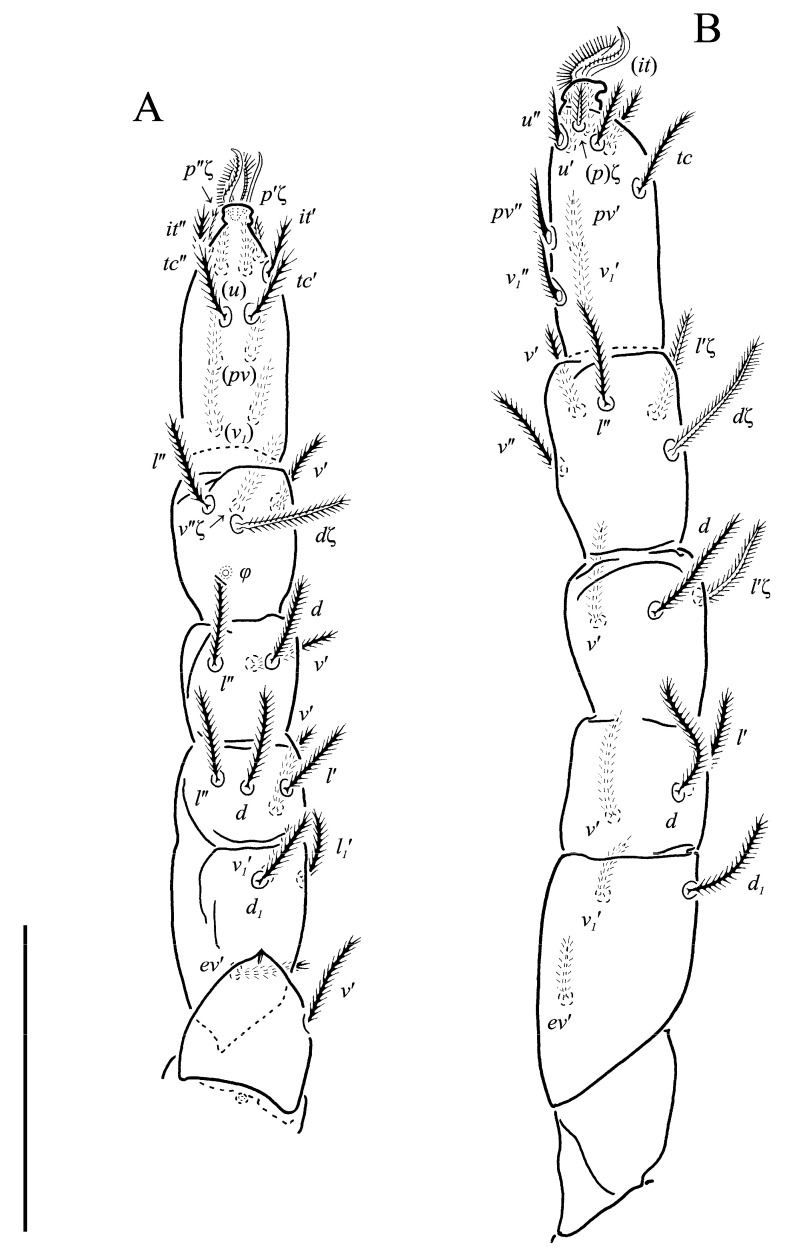
*Neoprotereunetes paulinae* (Gless, 1972), holotype female. (**A**) Left leg III, dorsal view; (**B**) left leg IV, lateral view. Scale bar: 50 μm.

**Figure 13 animals-13-02213-f013:**
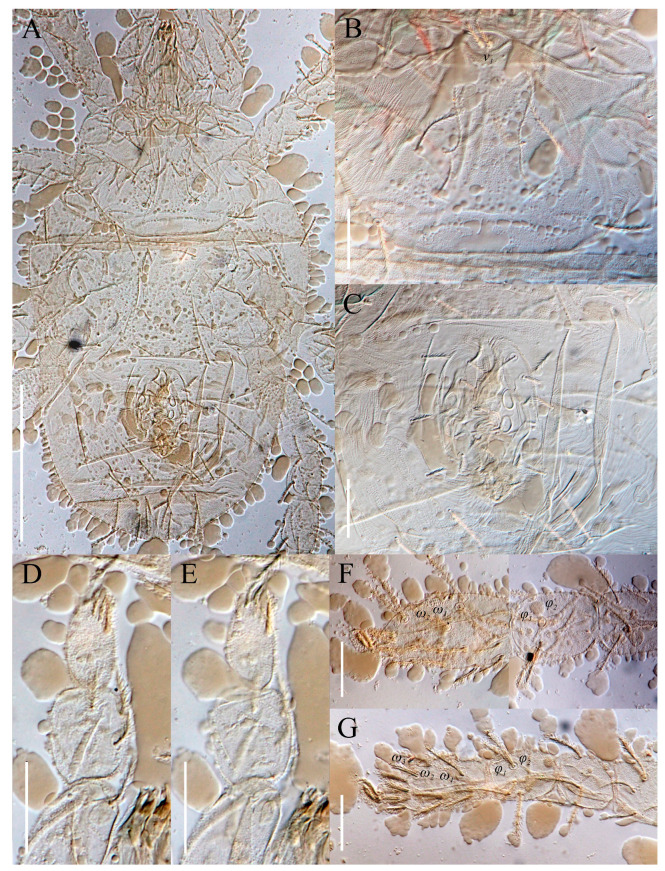
*Neoprotereunetes paulinae* (Gless, 1972), holotype female. (**A**) Body, dorsal and ventral view; (**B**) prodorsum; (**C**) genital region (**D**) tarsus and tibia of left palp, lateral view, antiaxial facesurface; (**E**) tarsus and tibia of left palp, lateral view, paraxial facesurface; (**F**) tarsus and tibia of right leg I, lateral view; (**G**) tarsus, tibia and genu of right leg II, ventrolateral view. Scale bar: (**A**) 100 μm; (**B**–**G**) 20 μm.

**Figure 14 animals-13-02213-f014:**
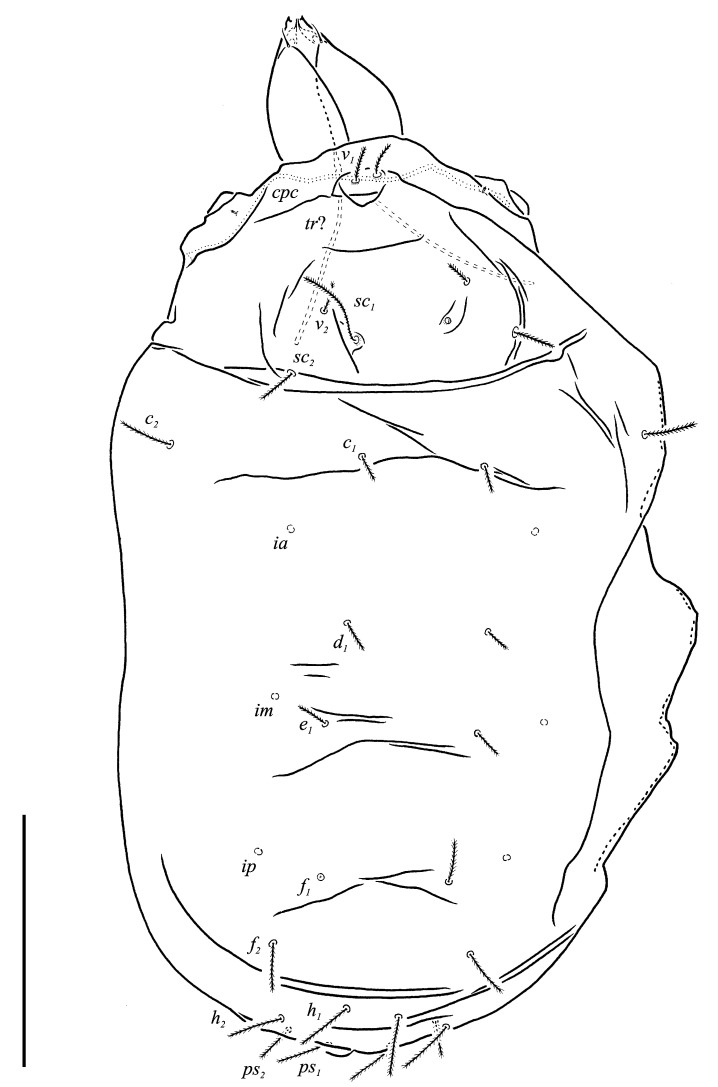
*Antarcteupodes maudae* (Strandtmann, 1967), holotype female. Body, dorsal view. Scale bar: 100 μm.

**Figure 15 animals-13-02213-f015:**
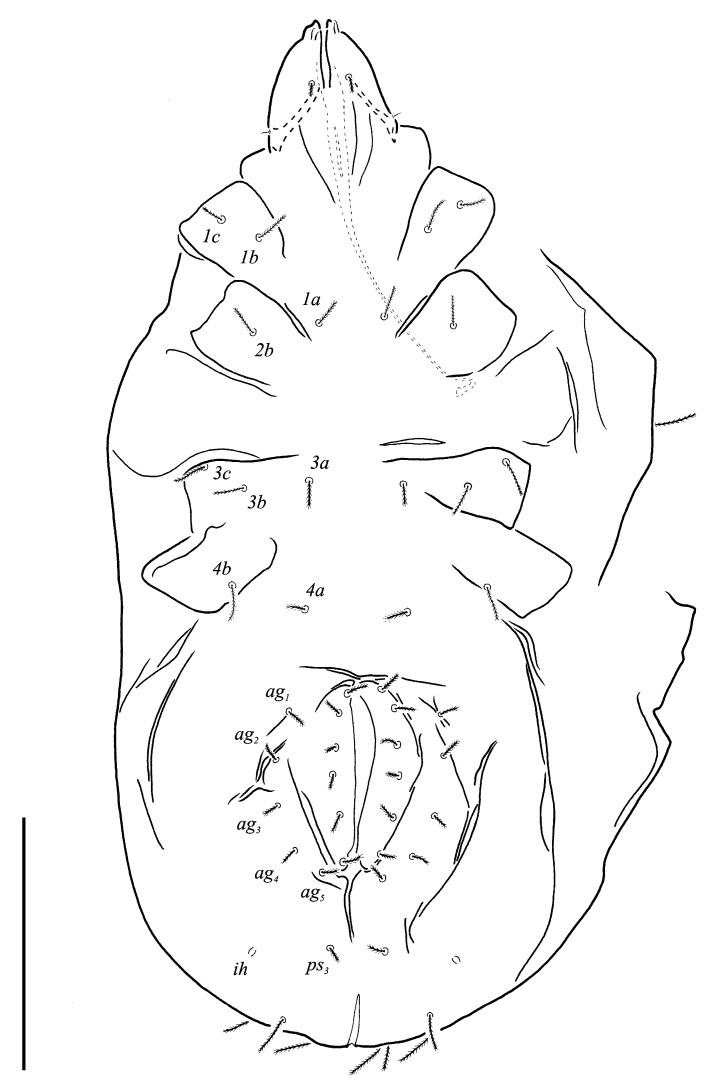
*Antarcteupodes maudae* (Strandtmann, 1967), holotype female. Body, ventral view. Scale bar: 100 μm.

**Figure 16 animals-13-02213-f016:**
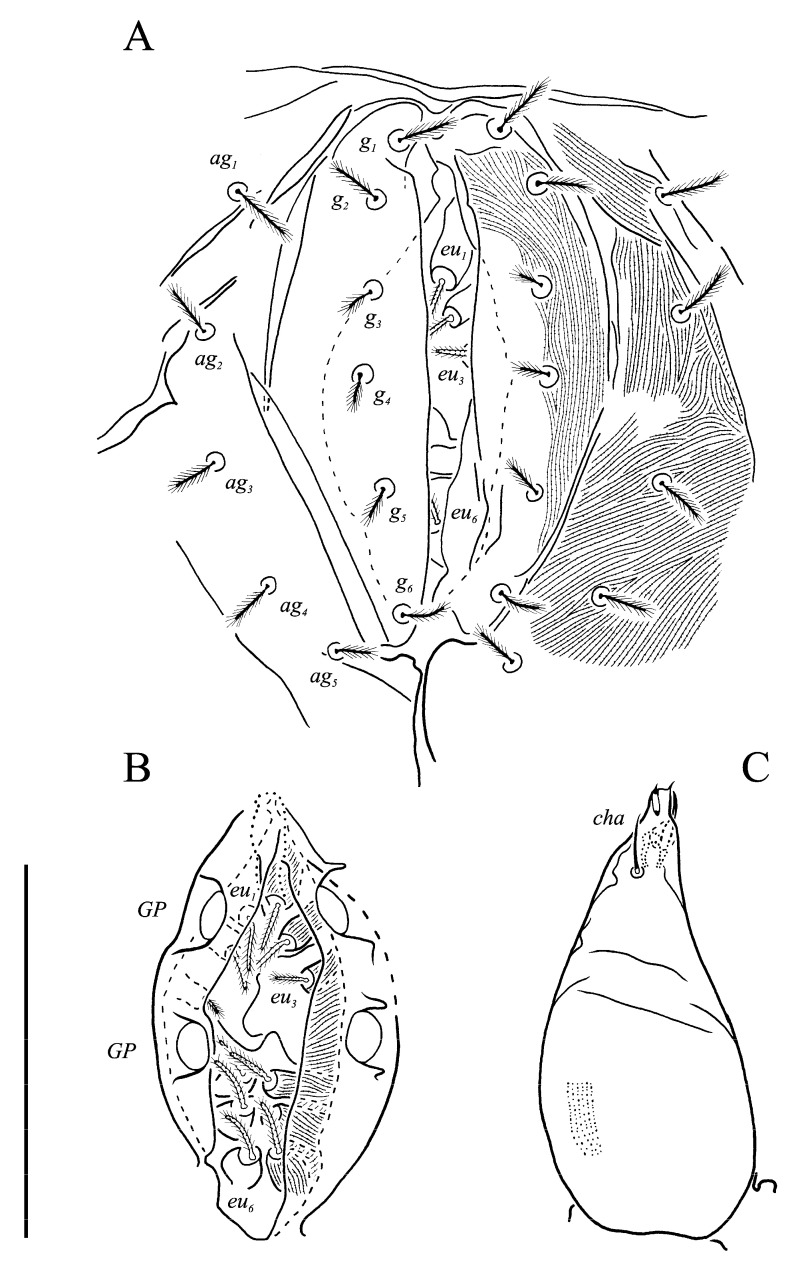
*Antarcteupodes maudae* (Strandtmann, 1967), holotype female. (**A**) Genital region; (**B**) progenital chamber; (**C**) left chelicera, dorsal view. Scale bar: 50 μm.

**Figure 17 animals-13-02213-f017:**
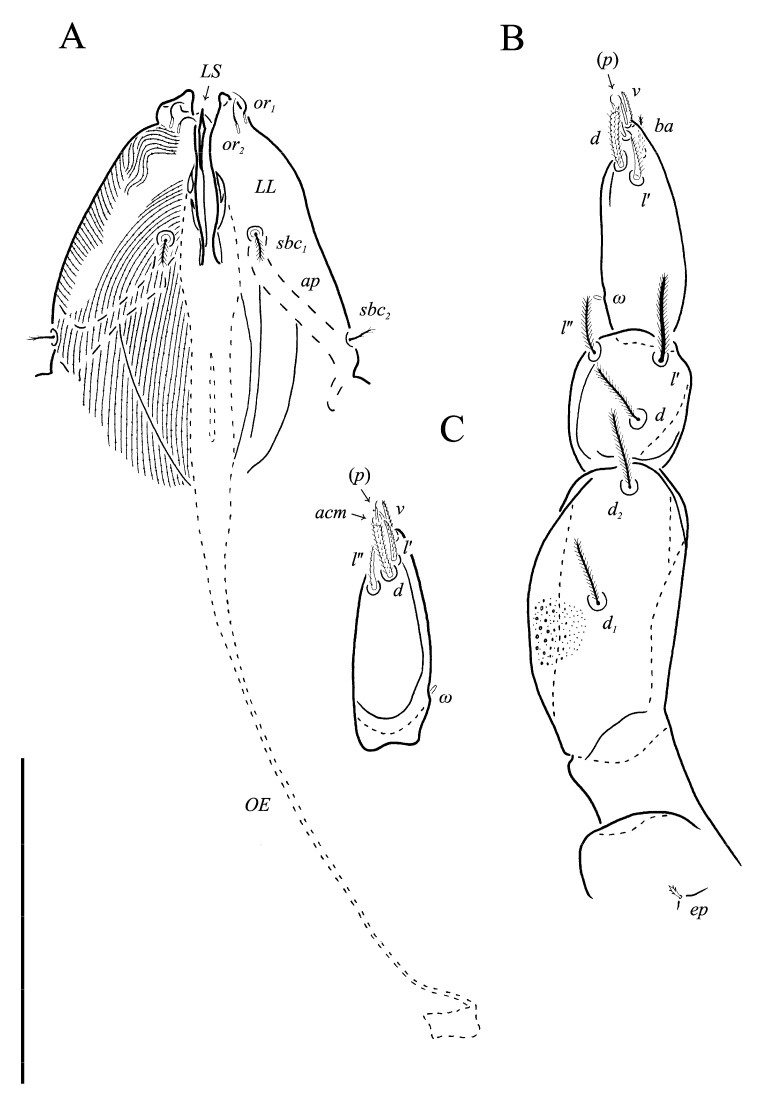
*Antarcteupodes maudae* (Strandtmann, 1967), holotype female. (**A**) Subcapitulum, ventral view; (**B**) left palp, dorsal view; (**C**) tarsus of right palp, dorsal view. Scale bar: 50 μm.

**Figure 18 animals-13-02213-f018:**
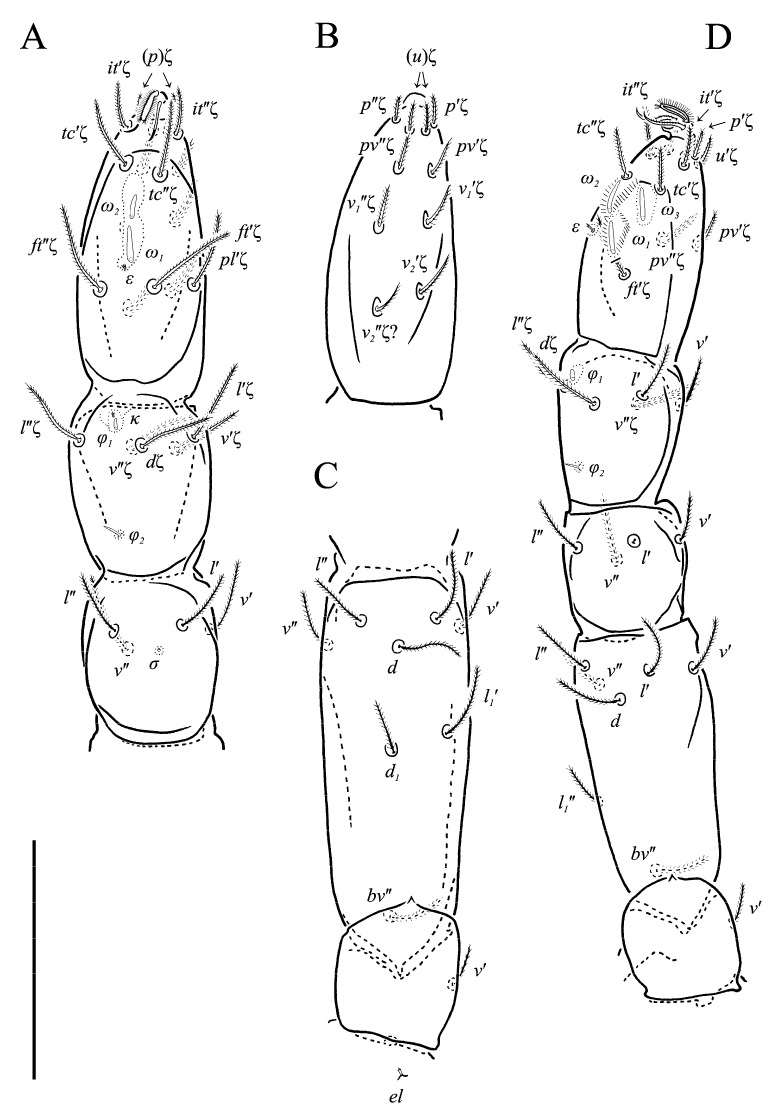
*Antarcteupodes maudae* (Strandtmann, 1967), holotype female. (**A**) Tarsus, tibia and genu of left leg I, dorsal view; (**B**) tarsus of left leg I, ventral view; (**C**) femur and trochanter of left leg I, dorsal view; (**D**) left leg II, dorsolateral view. Scale bar: 50 μm.

**Figure 19 animals-13-02213-f019:**
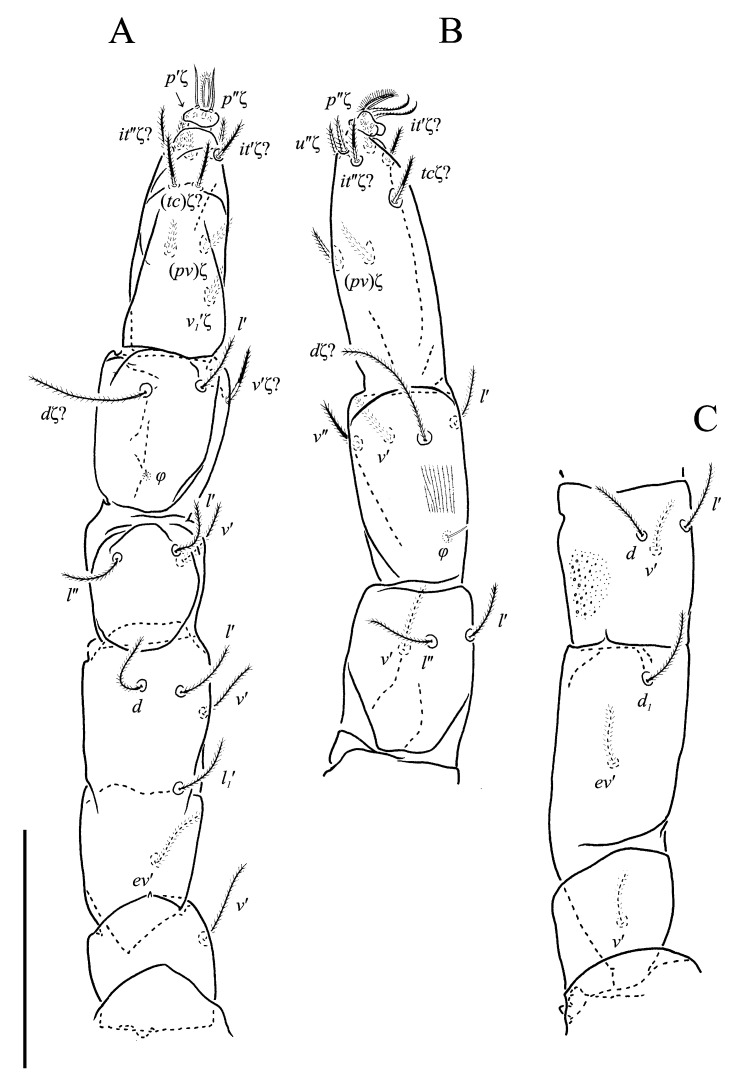
*Antarcteupodes maudae* (Strandtmann, 1967), holotype female. (**A**) Left leg III, dorsal view; (**B**) tarsus, tibia and genu of left leg IV, dorsolateral view; (**C**) telo-, basifemur and trochanter of left leg IV, dorsolateral view. Scale bar: 50 μm.

**Figure 20 animals-13-02213-f020:**
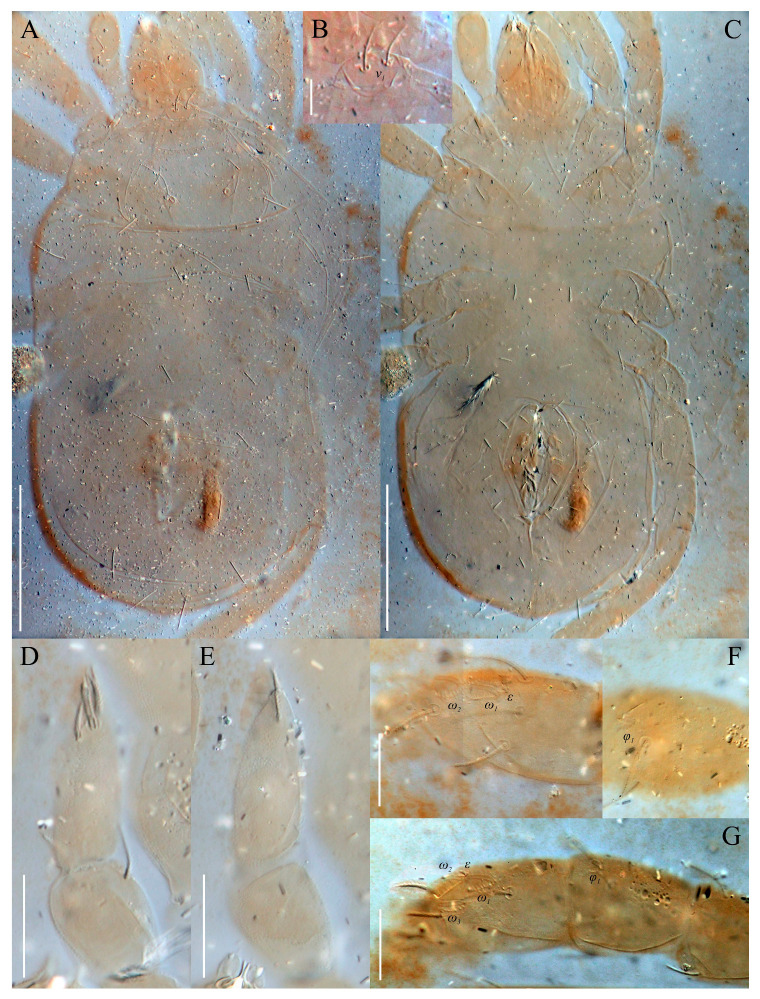
*Antarcteupodes maudae* (Strandtmann, 1967), holotype female. (**A**) Body, dorsal view; (B) naso; (**C**) body, ventral view; (**D**) tarsus and tibia of right palp, dorsal view; (**E**) tarsus and tibia of right palp, ventral view; (**F**) tarsus and tibia of right leg I, dorsolateral view; (**G**) tarsus and tibia of right leg II, dorsolateral view. Scale bar: (**A**,**C**) 100 μm; (**B**) 10 μm; (**D**–**G**) 20 μm.

**Figure 21 animals-13-02213-f021:**
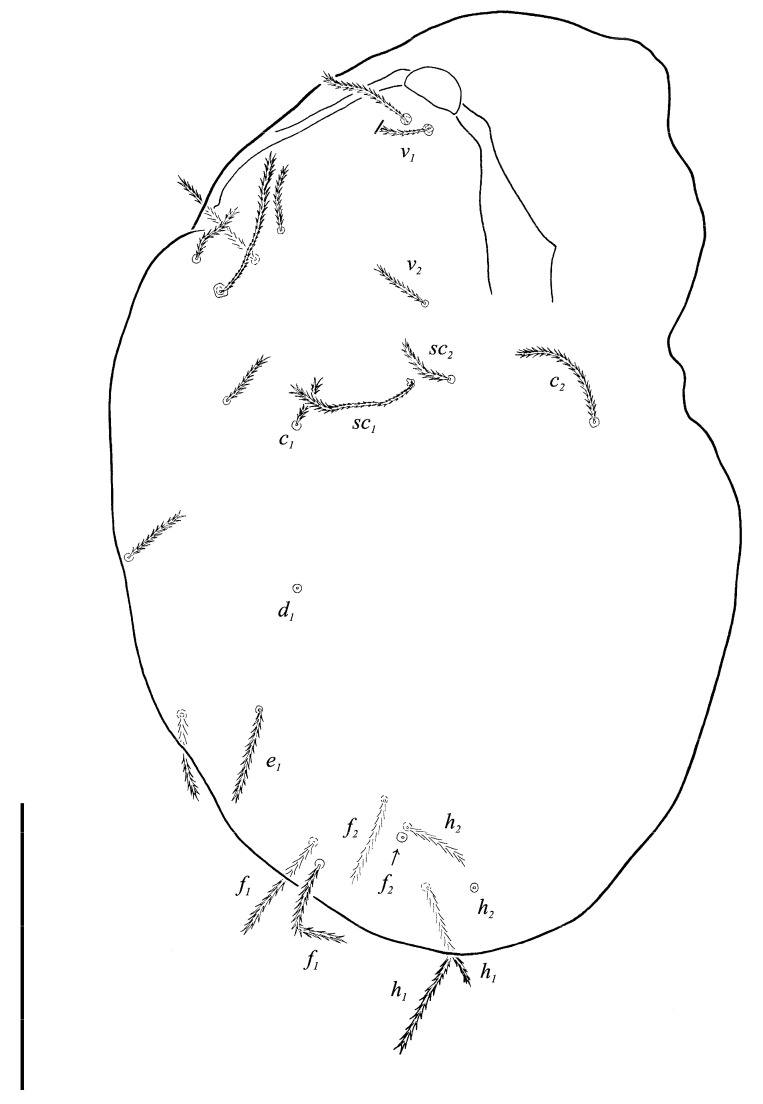
*Filieupodes lapidarius* (Oudemans, 1906), holotype female. Idiosoma, dorsolateral view. Scale bar: 100 μm.

**Figure 22 animals-13-02213-f022:**
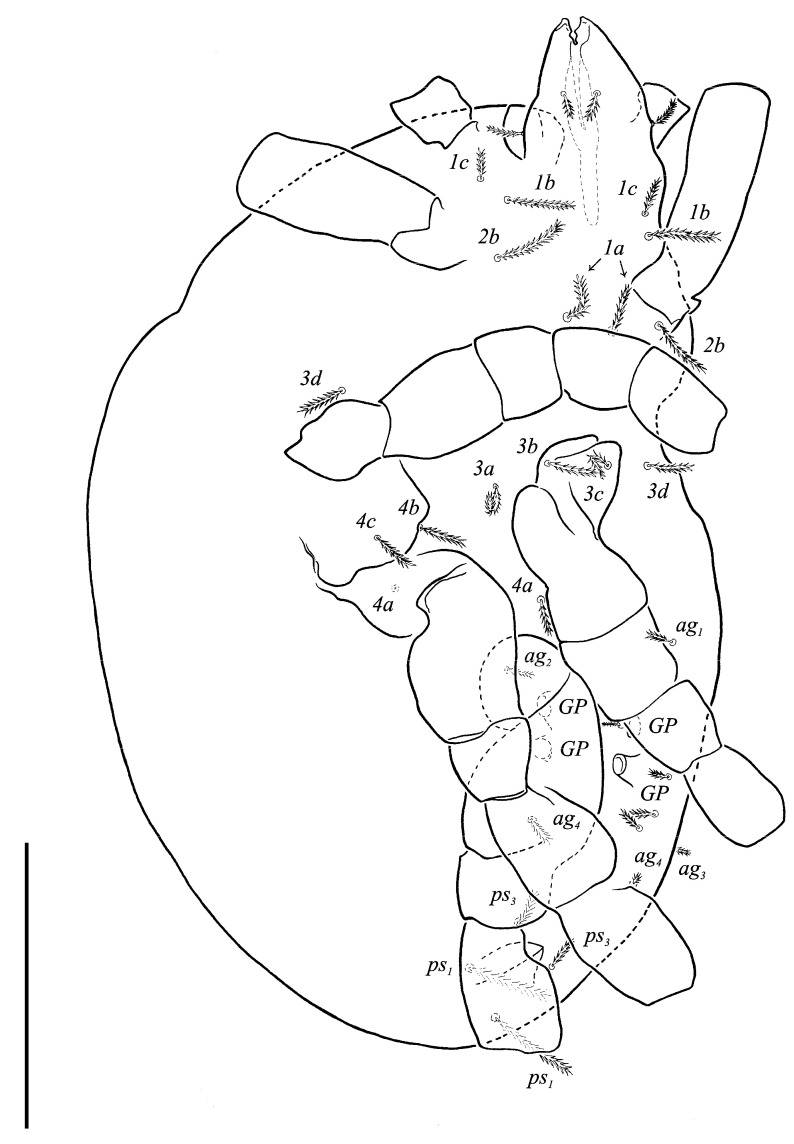
*Filieupodes lapidarius* (Oudemans, 1906), holotype female. Body, ventral view. Scale bar: 100 μm.

**Figure 23 animals-13-02213-f023:**
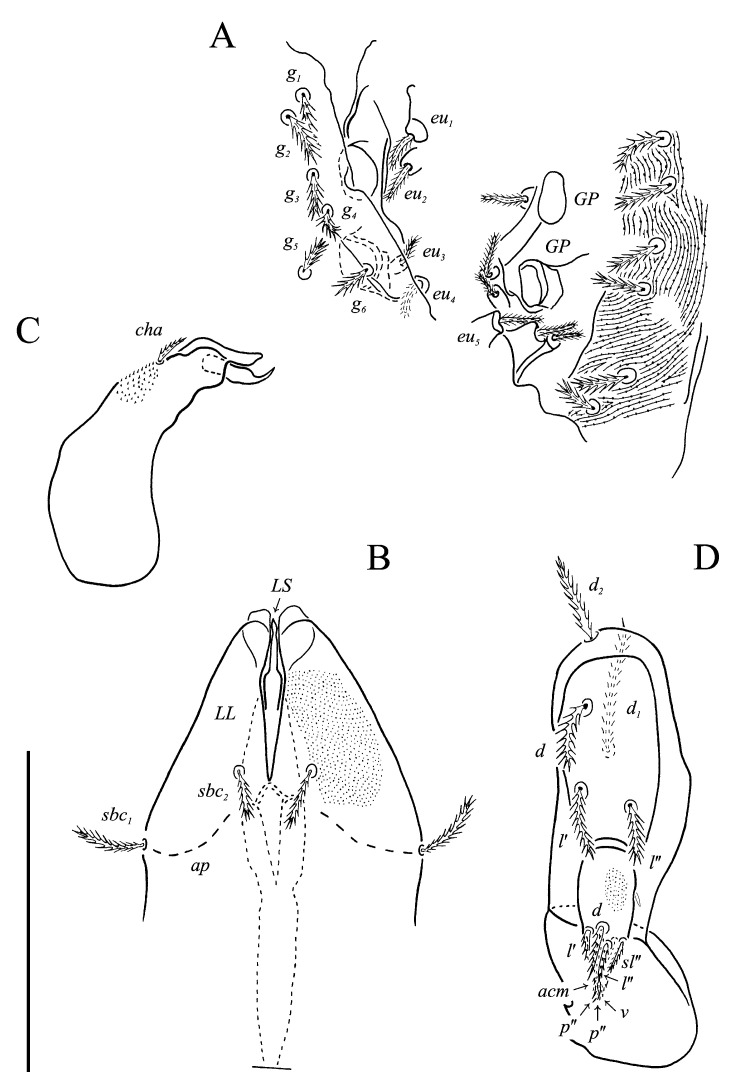
*Filieupodes lapidarius* (Oudemans, 1906), holotype female. (**A**) Genital region; (**B**) subcapitulum, ventral view; (**C**) left chelicera, lateral view; (**D**) left palp, dorsal view. Scale bar: 50 μm.

**Figure 24 animals-13-02213-f024:**
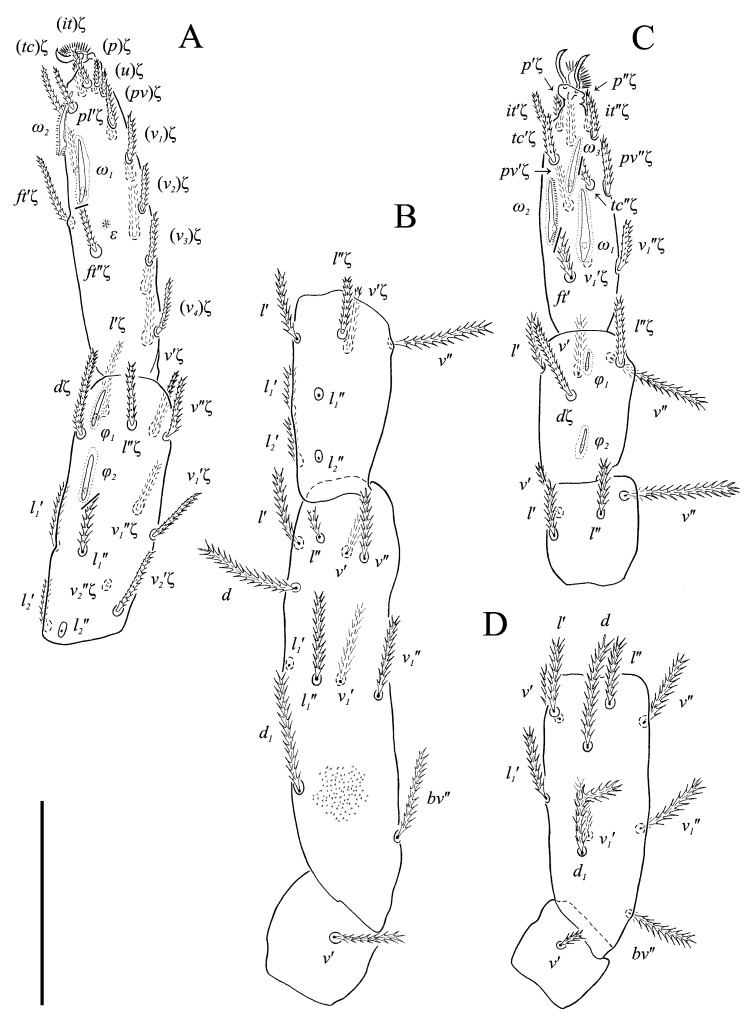
*Filieupodes lapidarius* (Oudemans, 1906), holotype female. (**A**) Tarsus and tibia of right leg I, dorsolateral view; (**B**) genu, femur and trochanter of right leg I, dorsolateral view; (**C**) tarsus, tibia and genu of right leg II, dorsal view; (**D**) femur and trochanter of right leg II, dorsal view. Scale bar: 50 μm.

**Figure 25 animals-13-02213-f025:**
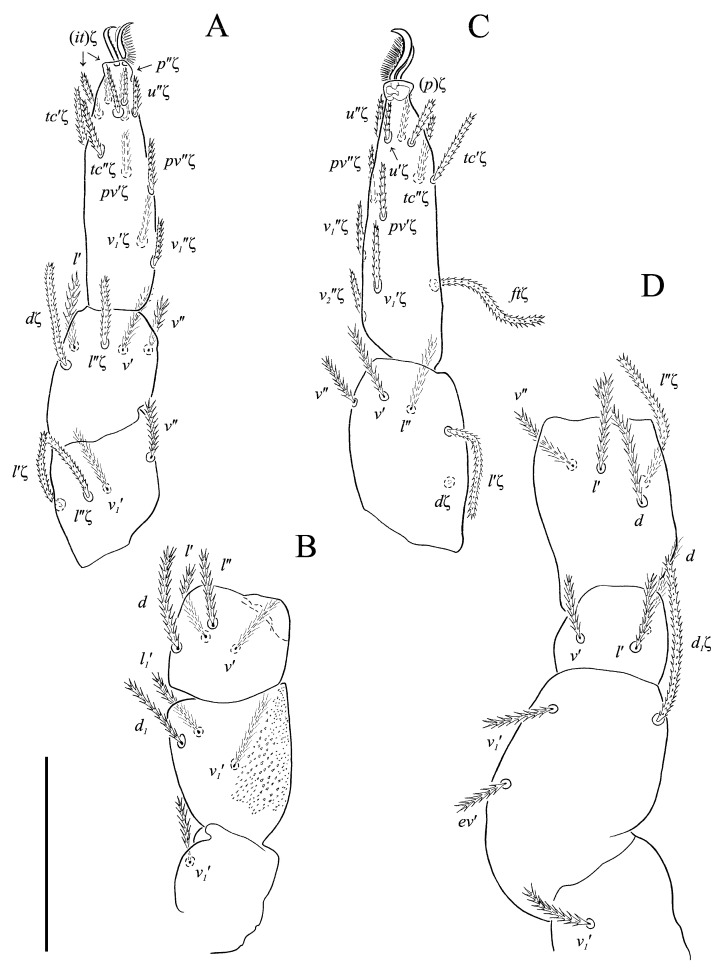
*Filieupodes lapidarius* (Oudemans, 1906), holotype female. (**A**) Tarsus, tibia and genu of right leg III, lateral view; (**B**) femur and trochanter of right leg III, lateral view; (**C**) tarsus and tibia of left leg IV, lateral view; (**D**) genu, femur and trochanter of left leg IV, lateral view. Scale bar: 50 μm.

**Figure 26 animals-13-02213-f026:**
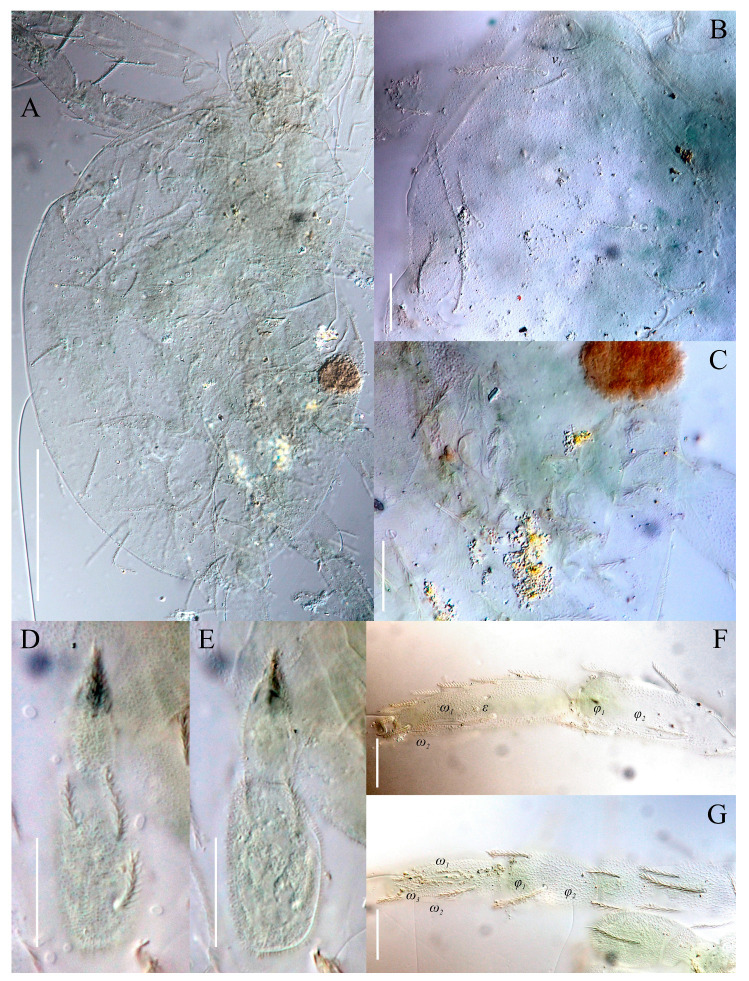
*Filieupodes lapidarius* (Oudemans, 1906), holotype female. (**A**) Body, dorsal view; (**B**) prodorsum; (**C**) genital region; (**D**) tarsus and tibia of right palp, dorsal view; (**E**) tarsus and tibia of right palp, ventral view; (**F**) tarsus and tibia of right leg I, lateral view; (**G**) tarsus, tibia and genu of right leg II, dorsal view. Scale bar: (**A**) 100 μm; (**B**–**G**) 20 μm.

## Data Availability

The data presented in this study are available on request from the corresponding author.

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
