# Peer review of "New Definition of Neoprotereunetes Fain et Camerik, Its Distribution and Description of the New Genus in Eupodidae (Acariformes: Prostigmata: Eupodoidea)â€"

_animals, 2023, doi:10.3390/ani13132213_

Round 1
Reviewer 1 Report
Dear Authors,
This is an excellent article adding value to the Eupodidae taxonomy. I made suggestions and corrections in the manuscript.

The English is good and I made a few corrections/suggestions.
Author Response
Response to Reviewer 1
All the corrections and suggestions are accepted and applied, including the one concerning designation of neotype.
Reviewer 2 Report
The submitted manuscript is an excellent and major contribution to the systematics of poorly knoen family Eupodidae. Illustrations of a very high quality as well as detailed descriptions of taxa.
Excellent work! I just made some minor corrections and notes in the manuscript.

Author Response
Response to Reviewer 2
p. 1 (Abstract) Comment “I agree that it is a distinct taxon (but probably not a family level), however it includes the genus Linopodes. The family name Linopodidae were mentioned many times before Cocceupodidae. Just enter Linopodidae in any browser. So, Cocceupodidae should be a junior synonym of Linopodidae.”
Though, we do agree that Cocceupodidae should be at most subfamily-level taxon, we did not change its original rank (for reasons see Szudarek-Trepto et al. 2020). We tried to find the original paper apllying family Linopodidae in taxonomic context, but we failed. There are few papers (e.g., Solomon 1937; Vistorin-Theis 1977; Ehrnsberger 1988) containing the name Linopodidae, but it is just mentioned in the text without author and year of description or any reference to original paper. Linopodidae might contain Linopodes only or be erroneously used as the synonym of Eupodidae. As diagnosis, authorship or systematic position of Linopodidae are not available, we cannot use it. Supplying that we know original paper with description of Linopodidae, we may consider the change of the name of Linopodes-Cocceupodes-Filieupodes clade taxon, provided that all the requirements for name availability (ICZN, Art. 13) are met, so the proper nomenclatural act can be performed.
p. 1 (Introduction) Comment “Also please mention that according to molecular phylogenetic reconstruction of Acariformes (Klimov et al. 2018) Rhagidiidae placed far from other eupodoid mites and probably not belongs to Eupodoidea.”
Indeed, there is still no agreement if Rhagidiidae really belongs to Eupodoidea. There are also some sound phylogenetic reconstructions where Rhagidiidae were recovered within Eupodoidea (e.g., Dabert et al. 2016, Szudarek-Trepto et al. 2023). As the matter of origin of Rhagidiidae (and also, e.g., problem of validity of Strandtmannidae and Dendrochetidae) are far beyond the scope of our study we would not like to get involved into that (complex) matter.
p. 3 (Material and methods) Comment “Why with question mark?”
As explained in the description of A. maudae, the identity of the reported structures is uncertain and hence the question mark after the abbreviation.
p. 4 (Results) Comment “Please also provide differences between Neoprotereunetes and Claveupodes.”
We think that the differential diagnosis of Neoprotereunetes is already quite elaborate and rather not to be extended more. Besides Claveupodes, there are other eupodid genera that resemble Neoproterenetes (Echinoeupodes, Niveupodes) and it is impossible to provide the differential diagnoses to all of them. Moreover, we know of yet another eupodid genus awaiting its description, that seems to be more closely related to Claveupodes and it will be compared to that genus in an upcoming paper. There probably will be publishesd also the new key to genera.
p. 5 (Results) Comment “Is it possible to illustrate the type of prodorsal ornamentation?”
Actually the spicules seem to create lines, but their course change visually when slightly turning the fine focus adjustment knob. They, however, can partially be seen in the Figure 12B.
p. 24 (Results; Antarcteupodes maudae) Comment “Also add lengths of erect solenidia.”
The lengths of erect solenidia are excluded from descriptions because these structures, unlike rhagidial organs, are well visible and possible to measure only when viewed laterally. Otherwise they are partially or completely situated along optical axis and their measurements would not be reliable.
All the remaining corrections and suggestions are accepted and applied.